# Fast and Provable Algorithms for Sparse PCA with Improved Sample Complexity

Jian-Feng Cai [1]   Zhuozhi Xian [1]   Jiaxi Ying [1]

## Abstract

We explore the single-spiked covariance model within the context of sparse principal component analysis (PCA), which aims to recover a sparse unit vector from noisy samples. From an information-theoretic perspective, $O(k \log p)$ observations are sufficient to recover a $k$-sparse $p$-dimensional vector $\boldsymbol{v}$. However, existing polynomial-time methods require at least $O(k^2)$ samples for successful recovery, highlighting a significant gap in sample efficiency. To bridge this gap, we introduce a novel thresholding-based algorithm that requires only $\Omega(k \log p)$ samples, provided the signal strength $\lambda = \Omega(\|\boldsymbol{v}\|_\infty^{-1})$. We also propose a two-stage nonconvex algorithm that further enhances estimation performance. This approach integrates our thresholding algorithm with truncated power iteration, achieving the minimax optimal rate of statistical error under the desired sample complexity. Numerical experiments validate the superior performance of our algorithms in terms of estimation accuracy and computational efficiency.

## 1. Introduction

Principal component analysis (PCA) is a cornerstone technique for dimension reduction, data preprocessing, clustering and feature extraction. However, in high–dimensional regimes, its dependence on all variables often leads to overfitting and poor interpretability, obscuring the truly important features (Johnstone & Lu, 2009). By introducing sparsity into principal components, sparse PCA can mitigate issues of overfitting and loss of interpretability that often arise in high-dimensional settings, particularly when the number of variables surpasses the number of samples (Zou

et al., 2006; Shen & Huang, 2008), which offers a robust alternative to traditional PCA.

In this paper, we investigate the single-spiked covariance model (Johnstone, 2001), which involves recovering a sparse vector $\boldsymbol{v}$ from $n$ noisy samples:

$$\boldsymbol{x}_i = \sqrt{\lambda} g_i \boldsymbol{v} + \boldsymbol{\xi}_i, \quad i = 1, \ldots, n, \qquad (1)$$

where $\boldsymbol{x}_i \in \mathbb{R}^p$ denotes the $i$-th observation and $\boldsymbol{v} \in \mathbb{R}^p$ is the unknown vector to be estimated, characterized by $\|\boldsymbol{v}\|_0 \leq k$ and $\|\boldsymbol{v}\|_2 = 1$. The coefficients $g_i$ are sampled independently from a standard normal distribution $\mathcal{N}(0, 1)$, and the noise vectors $\boldsymbol{\xi}_i$ are likewise independently drawn from a multivariate normal distribution $\mathcal{N}(\boldsymbol{0}, \boldsymbol{I})$, with each $g_i$ and $\boldsymbol{\xi}_i$ being mutually independent. The parameter $\lambda > 0$ quantifies the strength of the signal.

From an information-theoretic perspective, it has been shown that $n = \Omega(k \log p)$ samples are sufficient to estimate a $k$-sparse $p$-dimensional vector $\boldsymbol{v}$ up to a constant error (Vu & Lei, 2013; Berthet & Rigollet, 2013). However, achieving such an estimate are nonpolynomial-time algorithms such as exhaustive search. In contrast, all known polynomial-time algorithms, including diagonal thresholding (Johnstone & Lu, 2009), covariance thresholding (Krauthgamer et al., 2015; Deshpande & Montanari, 2016), and convex relaxation techniques (d'Aspremont et al., 2004; Amini & Wainwright, 2009; Ma & Wigderson, 2015), need at least $\Omega(k^2)$ observations to estimate $\boldsymbol{v}$. Moreover, the known lower bounds for these methods are considered tight (Krauthgamer et al., 2015; Ma & Wigderson, 2015; Deshpande & Montanari, 2016; Gao et al., 2017) when the signal strength $\lambda$ is treated as a constant.

Thus, for $k = \Omega(\log p)$, there exists a notable gap between the number of samples needed from an information-theoretic perspective, $\Omega(k \log p)$, and the number required by existing polynomial-time algorithms, $\Omega(k^2)$. Moreover, reductions from the planted-clique conjecture imply that, without further assumptions, no polynomial-time algorithm can attain the information-theoretic rate (Berthet & Rigollet, 2013; Krauthgamer et al., 2015; Wang et al., 2016; Gao et al., 2017; Brennan et al., 2018). This motivates the key question of our work:

[1]Department of Mathematics, Hong Kong University of Science and Technology, Hong Kong. Correspondence to: Jiaxi Ying <jx.ying@connect.ust.hk>.

*Proceedings of the 42^{nd} International Conference on Machine Learning*, Vancouver, Canada. PMLR 267, 2025. Copyright 2025 by the author(s).

*Can we design a polynomial-time algorithm to bridge this sample-complexity gap, under certain mild assumptions on the model?*

We answer this question affirmatively by designing a novel thresholding-based algorithm. We demonstrate that $\Omega(k \log p)$ is sufficient for our algorithm to estimate $\boldsymbol{v}$ up to a constant error (in euclidean norm), given that the signal strength $\lambda = \Omega(\|\boldsymbol{v}\|_\infty^{-1})$. By imposing a condition on $\lambda$ and the maximum absolute value of $\boldsymbol{v}$, we effectively reduce the parameter space of the model. Consequently, our algorithm achieves the desired sample complexity within this constrained parameter space. While the theoretical analysis of sparse PCA conditioned on the signal strength $\lambda$ has been studied in the literature (Johnstone & Lu, 2009; Deshpande & Montanari, 2016; Novikov, 2023; Ding et al., 2024), these studies predominantly focus on scenarios with relatively large sparsity $k$ and correspondingly high sample complexities, such as $n = \Omega(p)$. In contrast, our work aims to achieve a reduced sample complexity of $n = \Omega(k \log p)$, operating under a specific condition on $\lambda$.

Although the methods previously mentioned, such as diagonal thresholding (Johnstone & Lu, 2009), covariance thresholding (Krauthgamer et al., 2015; Deshpande & Montanari, 2016), and semidefinite programming (SDP) (d'Aspremont et al., 2004; Amini & Wainwright, 2009; Vu et al., 2013; Ma & Wigderson, 2015), are all computable in polynomial time, the SDP approach entails a significantly higher computational burden compared to thresholding-based methods. For instance, the SDP implementation by (d'Aspremont et al., 2004) incurs a computational complexity exceeding $O(p^4)$, rendering it intractable in the big data scenarios. Among the thresholding-based algorithms, covariance thresholding (Krauthgamer et al., 2015; Deshpande & Montanari, 2016) generally offers better estimation performance than diagonal thresholding (Johnstone & Lu, 2009). However, it requires eigendecomposition of a typically $p \times p$ matrix, costing $O(np^2 + p^3)$ totally, whereas diagonal thresholding involves eigendecomposition of a smaller $k \times k$ matrix, leading to a overall computational cost of $O(np + nk^2)$. In this paper, we aim to develop an algorithm that achieves both high estimation accuracy and computational efficiency.

## 1.1. Contributions

We introduce two complementary algorithms to address both sample complexity and estimation performance in the single-spiked covariance model. The first is a thresholding-based, polynomial-time algorithm that bridges the existing gap in sample complexity. The second is a two-stage nonconvex algorithm, which leverages our thresholding algorithm combined with truncated power iteration to further refine estimation. Our main contributions are:

- A notable gap exists between the information-theoretic sample limit of $\Omega(k \log p)$ to achieve a constant estimation error and the sample complexity of $\Omega(k^2)$ required by existing polynomial-time algorithms. We bridge this gap by introducing a novel polynomial-time algorithm based on the thresholding technique, which requires only $\Omega(k \log p)$ samples to achieve a constant error, provided the signal strength $\lambda = \Omega(\|\boldsymbol{v}\|_\infty^{-1})$.

  In existing work, the signal strength $\lambda$ is often treated as a constant when deriving both the information-theoretic lower bound of $\Omega(k \log p)$ and the sample complexities of standard algorithms (e.g., $\Omega(k^2 \log p)$ for diagonal thresholding and $\Omega(k^2)$ for covariance thresholding). Our algorithm achieves the optimal sample complexity of $\Omega(k \log p)$, provided that $\lambda = \Omega(\|\boldsymbol{v}\|_\infty^{-1})$. Under this condition, $\lambda$ remains constant provided that $\|\boldsymbol{v}\|_\infty = \Omega(1)$. This scenario is naturally satisfied by signals whose nonzero entries exhibit a power-law decay (Jagatap & Hegde, 2019). This property is known as compressibility in the compressive sensing literature (Donoho, 2006; Candes et al., 2006).

- To enhance estimation performance, we propose a two-stage nonconvex algorithm. The first stage uses our thresholding-based approach for initialization, followed by a refinement through truncated power iteration. We show that the total estimation error of our algorithm can be separated into two components: an optimization error, which decays at a linear rate, and a statistical error, which achieves the minimax optimal rate when $n = \Omega(k \log p)$ and $\lambda = \Omega(\|\boldsymbol{v}\|_\infty^{-1})$.

- Numerical experiments demonstrate that our proposed algorithm offers significant advantages in terms of both estimation accuracy and computational efficiency. In contrast, the diagonal thresholding method results in substantial estimation errors, while the two covariance thresholding approaches incur considerable computational costs.

**Notations:** $\|\boldsymbol{x}\|_0$ counts the nonzero entries of vector $\boldsymbol{x}$, while $\|\boldsymbol{x}\|_2$ and $\|\boldsymbol{x}\|_\infty$ denote its $\ell_2$ and $\ell_\infty$ norms respectively. $\boldsymbol{x}_{\mathcal{T}}$ represents the vector of the same length as $\boldsymbol{x}$, where elements indexed by $\mathcal{T}$ are retained, and elements indexed by $\mathcal{T}^c$ are set to zero. $[\boldsymbol{x}]_{\mathcal{T}}$ denotes the sub-vector of $\boldsymbol{x}$ containing only the elements indexed by $\mathcal{T}$. $\boldsymbol{A}_{\mathcal{T}}$ denotes the matrix of the same size as a square matrix $\boldsymbol{A}$, where columns and rows indexed by $\mathcal{T}$ are retained, and those indexed by $\mathcal{T}^c$ are set to zero. $[\boldsymbol{A}]_{\mathcal{T}}$ refers to the sub-matrix of $\boldsymbol{A}$ composed of the columns and rows indexed by $\mathcal{T}$. We write $f(n) = O(g(n))$ means $f(n) \leq c g(n)$ and $f(n) = \Omega(g(n))$ means $f(n) \geq c' g(n)$ for positive constants $c$ and $c'$, and use $[p]$ to denote the set $1, \ldots, p$.

## 2. Preliminaries and Related Work

In this section, we begin by introducing the preliminaries of sparse PCA, then present related work.

### 2.1. Preliminaries

The single-spiked covariance model, initially introduced by (Johnstone, 2001), has become a foundational framework for sparse PCA. It outlines a methodology to estimate a single sparse vector $\boldsymbol{v}$ from $n$ noisy observations, denoted as $\boldsymbol{x}_1, \ldots, \boldsymbol{x}_n \in \mathbb{R}^p$. Each noisy sample $\boldsymbol{x}_i$ is independently drawn as in (1) and is therefore zero-mean.

We compute the empirical covariance matrix as $\hat{\boldsymbol{\Sigma}} = \frac{1}{n} \sum_{i=1}^{n} \boldsymbol{x}_i \boldsymbol{x}_i^T$, and its expectation is given by (Ma & Wigderson, 2015; Krauthgamer et al., 2015):

$$\mathbb{E}\big[\hat{\boldsymbol{\Sigma}}\big] = \lambda \boldsymbol{v} \boldsymbol{v}^T + \boldsymbol{I}. \tag{2}$$

Given an empirical covariance matrix $\hat{\boldsymbol{\Sigma}} \in \mathbb{R}^{p \times p}$, the objective of sparse PCA is to identify a sparse unit vector $\boldsymbol{w} \in \mathbb{R}^p$ that maximizes the quadratic form $\boldsymbol{w}^\top \hat{\boldsymbol{\Sigma}} \boldsymbol{w}$. Formally, this can be expressed as:

$$\max_{\boldsymbol{w}} \boldsymbol{w}^T \hat{\boldsymbol{\Sigma}} \boldsymbol{w}, \quad \text{subject to } \|\boldsymbol{w}\|_2 = 1, \ \|\boldsymbol{w}\|_0 \le k. \tag{3}$$

Although the solution to (3) yields an estimator for the spike vector $\boldsymbol{v}$, its combinatorial $\ell_0$-norm constraint renders the problem NP-hard. In this paper, we address these challenges by proposing a polynomial-time algorithm that bridges the existing sample-complexity gap and a computationally efficient nonconvex algorithm that achieves the minimax optimal rate under the desired sample complexity.

### 2.2. Related Work

We review related work, specifically thresholding-based algorithms (Johnstone & Lu, 2009; Krauthgamer et al., 2015; Deshpande & Montanari, 2016) and convex relaxation techniques (d'Aspremont et al., 2004; Ma & Wigderson, 2015), which have made significant strides in solving the sparse PCA optimization problem.

Thresholding-based algorithms for sparse PCA have been widely adopted due to their simplicity and effectiveness. These algorithms operate by initially estimating the support of $\boldsymbol{v}$ and subsequently computing the principal eigenvector of the corresponding submatrix of $\hat{\boldsymbol{\Sigma}}$ to approximate $\boldsymbol{v}$. Notable among these are diagonal thresholding (Johnstone & Lu, 2009) and covariance thresholding algorithms (Krauthgamer et al., 2015; Deshpande & Montanari, 2016). Diagonal thresholding leverages the indices of the top $k$ diagonal entries of the empirical covariance matrix $\hat{\boldsymbol{\Sigma}}$ as an estimate of the support of $\boldsymbol{v}$. With $O(np + nk^2)$ computational complexity, its sample complexity is

$\Omega(k^2 \log p)$ (Amini & Wainwright, 2009), which is worse than the information theoretical bound $\Omega(k \log p)$. To attain lower sample complexity, covariance thresholding, in contrast, involves a more intricate process. It applies either a soft thresholding (Deshpande & Montanari, 2016) or hard thresholding (Krauthgamer et al., 2015) to the empirical covariance matrix $\hat{\boldsymbol{\Sigma}}$, followed by using the indices of the top $k$ entries of its principal eigenvector to estimate the support of $\boldsymbol{v}$. It has been proved that the sample complexity of covariance thresholding proposed by (Deshpande & Montanari, 2016) is $\Omega(k^2)$, but its computational complexity is $O(np^2 + p^3)$ as analysed in Section 1.

Convex relaxation methods, such as SDP relaxation (d'Aspremont et al., 2004) and degree-4 Sum-of-Squares (SoS) algorithm (Ma & Wigderson, 2015), offer an alternative approach. They reframe the non-convex sparse PCA optimization problem as a convex optimization problem, so it can be solved in polynomial time. SDP relaxation rewrites Problem (3) by introducing a new variable $\boldsymbol{W} = \boldsymbol{w} \boldsymbol{w}^T$ and then relaxes the $\ell_0$-norm constraint by a convex constraint. Its sample complexity is $\Omega(k^2 \log p)$ (Amini & Wainwright, 2009) and its computational complexity exceeds $O(p^4)$ (d'Aspremont et al., 2004). The degree-4 SoS algorithm applies the well-established SoS method (Lasserre, 2015) to the sparse PCA optimization problem, which requires larger computational costs than SDP but proves to be unable to improve its sample complexity (Ma & Wigderson, 2015).

## 3. Main Results

In this section, we begin by introducing two algorithms in Section 3.1. Following this, we examine their computational complexity in Section 3.2. Finally, in Section 3.3, we present a detailed theoretical analysis of both algorithms. This analysis includes establishing the sample complexity required for the thresholding algorithm to achieve a constant error, thereby bridging the existing gap regarding the sample complexity. We also provide bounds for the estimation error of the two-stage algorithm, detailing how it enhances performance through a refined estimation process.

### 3.1. Proposed Algorithms

The first algorithm is a thresholding-based algorithm, an advanced version of diagonal thresholding. The second algorithm is a nonconvex two-stage algorithm for enhancing the estimation performance.

#### 3.1.1. THRESHOLDING-BASED ALGORITHM

Our proposed algorithm extends the well-known diagonal thresholding method (Johnstone & Lu, 2009), which, while being one of the most computationally efficient approaches,

suffers from a high sample complexity of $\Omega(k^2 \log p)$ (Amini & Wainwright, 2009). This complexity exceeds the information-theoretic lower bound of $\Omega(k \log p)$. To address this discrepancy, we first delve into the operational principles of diagonal thresholding.

Diagonal thresholding traditionally estimates the true spike by identifying the support and magnitude of the non-zero entries separately. This method estimates the support of $v$ by using the indices of the top $k$ diagonal entries of $\hat{\Sigma}$ in absolute value. This approach is based on the observation that the expectation values of the diagonal entries of $\hat{\Sigma}$ are given by:

$$\left[\mathbb{E}[\hat{\Sigma}]\right]_{jj} = \begin{cases} \lambda v_j^2 + 1, & j \in \mathcal{S}, \\ 1, & j \in \mathcal{S}^c, \end{cases} \tag{4}$$

where $\mathcal{S}$ denotes the support of $v$. There exists a gap between diagonal entries of the expectation in the sets $\mathcal{S}$ and $\mathcal{S}^c$. This statistical gap is defined by:

$$g_{\mathrm{d}} := \min_{j \in \mathcal{S}} \left| \left[\mathbb{E}[\hat{\Sigma}]\right]_{jj} \right| - \max_{j \in \mathcal{S}^c} \left| \left[\mathbb{E}[\hat{\Sigma}]\right]_{jj} \right| = \lambda \cdot \min_{j \in \mathcal{S}} v_j^2. \tag{5}$$

A simple analysis for diagonal thresholding is based on the following proposition about the statistical gap $g_{\mathrm{d}}$.

**Proposition 3.1.** *If $\left|\hat{\Sigma}_{ii} - \left[\mathbb{E}[\hat{\Sigma}]\right]_{ii}\right| \leq \frac{1}{2}g_d$ holds for every $i \in [p]$, then, for any $j \in \mathcal{S}$ and any $j' \in \mathcal{S}^c$, we have:*

$$\left|\hat{\Sigma}_{jj}\right| \geq \left|\hat{\Sigma}_{j'j'}\right|.$$

The proof of Proposition 3.1 is detailed in Appendix A.2. Given sufficient samples, the concentration error $\left|\hat{\Sigma}_{ii} - \left[\mathbb{E}[\hat{\Sigma}]\right]_{ii}\right|$ is small enough and the condition in Proposition 3.1 holds, hence the diagonal entries $\hat{\Sigma}_{jj}$ for indices $j \in \mathcal{S}$ are larger than those for indices $j \in \mathcal{S}^c$ in absolute values. Consequently, diagonal thresholding can estimate the support of $v$ by selecting the indices of the top $k$ diagonal entries of $\hat{\Sigma}$ in absolute value. Additionally, the sample complexity of diagonal thresholding is governed by the statistical gap $g_{\mathrm{d}}$. A larger gap $g_{\mathrm{d}}$ permits greater tolerance for the concentration error $\left|\hat{\Sigma}_{ii} - \left[\mathbb{E}[\hat{\Sigma}]\right]_{ii}\right|$, thereby reducing the number of samples required.

Based on the above observations, we aim to distinguish between two groups of entries corresponding to the sets $\mathcal{S}$ and $\mathcal{S}^c$, aiming for a larger gap between them. Our proposed thresholding algorithm draws inspiration from spectral initialization methods designed for sparse phase retrieval (Wu & Rebeschini, 2021; Cai et al., 2023). Instead of focusing on the diagonal entries, we examine the magnitudes of the entries in a specific column of $\hat{\Sigma}$. More precisely, we identify the support of $v$ by selecting the indices of the $k$ largest entries in the $j_0$-th column of $\hat{\Sigma}$ in absolute value, where $j_0$ is the index of the maximum element among the diagonal entries of $\hat{\Sigma}$.

Our method is motivated by the fact that, when $l \in \mathcal{S}$, the expected values of the entries in the $l$-th column of $\hat{\Sigma}$ can be expressed as:

$$\left[\mathbb{E}[\hat{\Sigma}e_l]\right]_j = \begin{cases} \lambda v_l^2 + 1, & j = l, \\ \lambda v_l v_j, & j \in \mathcal{S} \setminus \{l\}, \\ 0, & j \in \mathcal{S}^c, \end{cases} \tag{6}$$

where $\{e_j\}_{j=1}^n$ denotes the set of standard basis vectors in $\mathbb{R}^n$. Similar to (5) for diagonal thresholding, we now consider the following statistical gap:

$$\begin{aligned} g_c &:= \min_{j \in \mathcal{S}} \left|\left[\mathbb{E}[\hat{\Sigma}e_l]\right]_j\right| - \max_{j \in \mathcal{S}^c} \left|\left[\mathbb{E}[\hat{\Sigma}e_l]\right]_j\right| \\ &\geq \lambda |v_l| \cdot \min_{j \in \mathcal{S}} |v_j|. \end{aligned} \tag{7}$$

Upon comparison, we observe that the lower bound of the statistical gap in (7) consistently exceeds that in (5) when $l \in \mathcal{S}$. This indicates that our method more effectively distinguishes entries in $\mathcal{S}$ from those in $\mathcal{S}^c$ than diagonal thresholding, thereby suggesting a reduced sample complexity requirement (see Section 3.3 for details).

To maximize this lower bound, $l$ should ideally be selected as the index of the largest absolute element of $v$, which aligns with the index of the largest diagonal entry of $\mathbb{E}[\hat{\Sigma}]$ as shown in (2). However, since we only have the sample covariance matrix $\hat{\Sigma}$ rather than the population covariance $\mathbb{E}[\hat{\Sigma}]$, we designate $l$ as the index of the largest diagonal entry of $\hat{\Sigma}$, denoted by $j_0$.

Our thresholding-based algorithm is summarized in Algorithm 1 and consists of two main steps. First, we estimate the support of the spike vector $v$ by selecting the indices corresponding to the $k$ largest absolute values in $\hat{\Sigma}e_{j_0}$, which we denote as $\hat{\mathcal{S}}$. Second, we estimate $v$ via a procedure analogous to diagonal thresholding: we perform an eigendecomposition on the $k \times k$ submatrix $[\hat{\Sigma}]_{\hat{\mathcal{S}}}$ formed by restricting to rows and columns indexed by $\hat{\mathcal{S}}$.

The support-estimation step is critical for achieving the optimal $\Omega(k \log p)$ sample complexity. By amplifying the separation between in-support and out-of-support entries relative to diagonal thresholding, our method recovers the true support with fewer samples. This enhanced separation constitutes the key innovation of our approach. As an alternative spike-estimation step, one may simply normalize the $j_0$-th column of $[\hat{\Sigma} - I]_{\hat{\mathcal{S}}}$. This variant avoids the eigendecomposition for faster runtime, at the expense of a modest loss in estimation accuracy, yet it still achieves the optimal $\Omega(k \log p)$ sample complexity.

### 3.1.2. TWO-STAGE ALGORITHM

To enhance the estimation performance of our thresholding algorithm presented in Section 3.1.1, we propose a two-stage

---

**Algorithm 1** Thresholding-based algorithm for bridging the sample-complexity gap

---

**Input:** Samples $\{\boldsymbol{x}_i\}_{i=1}^n$, the sparsity $k$

Compute $\{\hat{\boldsymbol{\Sigma}}_{j,j}\}_{j=1}^n$ and set $j_0 = \arg\max_{1 \leq j \leq n} \hat{\boldsymbol{\Sigma}}_{j,j}$.

Compute $\hat{\boldsymbol{\Sigma}}\boldsymbol{e}_{j_0}$ and set $\hat{\mathcal{S}}$ as the indices of the top $k$ elements of $\hat{\boldsymbol{\Sigma}}\boldsymbol{e}_{j_0}$ in absolute value.

Compute $[\hat{\boldsymbol{\Sigma}}]_{\hat{\mathcal{S}}}$, set $[\boldsymbol{v^0}]_{\hat{\mathcal{S}}}$ as the unit leading eigenvector of $[\hat{\boldsymbol{\Sigma}}]_{\hat{\mathcal{S}}}$ and set $[\boldsymbol{v^0}]_{\hat{\mathcal{S}}^c} = 0$.

**Output:** $\boldsymbol{v}^0$

---

algorithm comprising both initialization and refinement stages. In the initialization stage, we utilize our thresholding algorithm to obtain an initial estimate. During the refinement stage, the estimate is refined by employing a well-known method called truncated power iteration (Yuan & Zhang, 2013), which is widely used for finding a sparse eigenvector of a matrix. By combining these two stages, our algorithm aims to achieve superior performance in addressing Problem (3).

The truncated power method (Yuan & Zhang, 2013) extends the classic power iteration—commonly used to extract a matrix's leading eigenvector—by inserting a sparsity-enforcing truncation at each step. For reference, the standard power updates are

$$\widetilde{\boldsymbol{v}}^t = \hat{\boldsymbol{\Sigma}}\boldsymbol{v}^{t-1}, \quad \boldsymbol{v}^t = \frac{\widetilde{\boldsymbol{v}}^t}{\|\widetilde{\boldsymbol{v}}^t\|_2}, \tag{8}$$

where $\widetilde{\boldsymbol{v}}^t$ is the intermediate update at $t$-th iteration and $\boldsymbol{v}^t$ its unit-norm normalization. The iteration (8) typically converges to a dense vector under suitable conditions. However, since the sparse PCA problem (3) seeks a sparse principal eigenvector, we employ the truncated power method instead of the traditional power method. The truncated power method modifies each iteration by truncating to zero all but the top $k'$ entries in terms of absolute value:

$$\widetilde{\boldsymbol{v}}^t = T_{k'}(\hat{\boldsymbol{\Sigma}}\boldsymbol{v}^{t-1}), \quad \boldsymbol{v}^t = \frac{\widetilde{\boldsymbol{v}}^t}{\|\widetilde{\boldsymbol{v}}^t\|_2}, \tag{9}$$

where $T_{k'}(\cdot)$ represents the truncation operator, which retains only the top $k'$ entries of a vector in absolute values and sets all others to zero. This modification makes the truncated power method a natural choice for Problem (3), as it inherently promotes sparsity in the resulting vector. It is common to set the truncation parameter $k'$ in the same order as the sparsity $k$ of the true spike. Moreover, since $\hat{\boldsymbol{\Sigma}} = \frac{1}{n}\sum_{i=1}^n \boldsymbol{x}_i\boldsymbol{x}_i^T$, $\hat{\boldsymbol{\Sigma}}\boldsymbol{v}^{t-1}$ in (9) can be simplified as $\frac{1}{n}\sum_{i=1}^n (\boldsymbol{x}_i^T\boldsymbol{v}^{t-1})\boldsymbol{x}_i$.

The convergence of truncated power iteration was established in (Yuan & Zhang, 2013), where it was shown to achieve the optimal statistical rate. However, it hinges on the assumption that the initial estimate $\boldsymbol{v}^0$ satisfies $|\sin\angle(\boldsymbol{v}^0, \boldsymbol{v})| \leq 1 - C$ with $C \in (0,1)$ a constant. Such an initial point was not provided in (Yuan & Zhang, 2013).

Our proposed thresholding algorithm introduced in Section 3.1.1 tackles this problem perfectly, which achieves the desired sample complexity. Building on this foundation, we propose a two-stage algorithm to solve Problem (3). The full algorithm is summarized in Algorithm 2.

---

**Algorithm 2** Two-stage algorithm for enhancing estimation

---

**Input:** Samples $\{\boldsymbol{x}_i\}_{i=1}^n$, the sparsity $k$, parameter $k'$

*// Initialization stage:*

Compute an initial estimate $\boldsymbol{v}^0$ by Algorithm 1.

*// Refinement stage:*

**for** $t = 1, 2, \ldots$ **do**
  $\widetilde{\boldsymbol{v}}^t = T_{k'}(\frac{1}{n}\sum_{i=1}^n (\boldsymbol{x}_i^T\boldsymbol{v}^{t-1})\boldsymbol{x}_i)$
  $\boldsymbol{v}^t = \widetilde{\boldsymbol{v}}^t/\|\widetilde{\boldsymbol{v}}^t\|_2$
**end for**
**Output:** $\boldsymbol{v}^t$

---

### 3.2. Computational Costs

The computational expenses associated with our thresholding algorithm, as proposed in Section 3.1.1, include: Computing the diagonal of $\hat{\boldsymbol{\Sigma}}$ requires $O(np)$ operations; Identifying the largest diagonal entry involves $O(p)$ operations; Computing $\hat{\boldsymbol{\Sigma}}\boldsymbol{e}_{j_0}$ requires $O(np)$ operations; Finding the indices of the top $k$ elements of $\hat{\boldsymbol{\Sigma}}\boldsymbol{e}_{j_0}$ in absolute value requires $O(p\log p)$ operations; Computing the submatrix $[\hat{\boldsymbol{\Sigma}}]_{\hat{\mathcal{S}}}$ requires $O(nk^2)$ operations; Computing the leading eigenvector of $[\hat{\boldsymbol{\Sigma}}]_{\hat{\mathcal{S}}}$ demands $O(k^3)$ operations. Combining these, the overall computational complexity of our thresholding algorithm totals $O(np + nk^2)$.

Remarkably, this computational complexity is equivalent to that of diagonal thresholding (Johnstone & Lu, 2009), yet significantly less than that of covariance thresholding (Krauthgamer et al., 2015; Deshpande & Montanari, 2016), which scales as $O(np^2 + p^3)$. Moreover, we will show in Section 3.3 that our approach offers improved sample complexity compared to both diagonal thresholding and covariance thresholding.

The two-stage algorithm employs our thresholding algorithm, costing $O(np + nk^2)$, and uses the truncated power iterations with $O(np)$ in each iteration. Given the limited number of iterations typically required for convergence, the overall computational expense of the refinement stage does not significantly increase the total cost of the algorithm.

### 3.3. Theoretical Results

In this section, we delve into the theoretical underpinnings of both the thresholding algorithm and the two-stage algorithm. A critical aspect of assessing these algorithms' effectiveness is quantifying the estimation error. To this end, we define a loss function that captures the error between the true vector $v$ and its estimate $\hat{v}$:

$$\text{dist}(v, \hat{v}) := \min \left\{ \|v - \hat{v}\|_2, \|v + \hat{v}\|_2 \right\}. \quad (10)$$

This loss function is chosen for the sign ambiguity of $\hat{v}$. Specifically, if $\hat{v}$ is a solution of the sparse PCA problem (3), $-\hat{v}$ is also a solution of (3), so $v$ can only be estimated up to a sign.

**Theorem 3.2** (Thresholding algorithm). *Let $v \in \mathbb{R}^p$ be a $k$-sparse unit vector. Let $x_i = \sqrt{\lambda} g_i v + \xi_i$, $i = 1, \ldots, n$, where $\lambda > 0$, $g_i \overset{\text{iid}}{\sim} \mathcal{N}(0, 1)$ and $\xi_i \overset{\text{iid}}{\sim} \mathcal{N}(0, I_p)$ are independent. For any $\gamma \in (0, 1]$, there exists universal constants $C_1, C_2 > 0$ such that if $\lambda \geq C_1\|v\|_\infty^{-1}$ and $n \geq C_2 \gamma^{-2} k \log p$, then with probability exceeding $1 - 5p^{-1}$, the output $v^0$ of Algorithm 1 satisfies $\text{dist}(v, v^0) \leq \gamma$.*

The proof of Theorem 3.2 is detailed in Appendix A.3. Theorem 3.2 demonstrates that our proposed algorithm requires only $n = \Omega(k \log p)$ samples to estimate the true spike $v$ up to constant error, given that the signal strength $\lambda = \Omega(\|v\|_\infty^{-1})$. This result is significant as it effectively bridges the gap between the information-theoretic lower bound of $\Omega(k \log p)$, necessary to achieve a constant estimation error, and the considerably higher sample complexity of $\Omega(k^2)$ demanded by existing polynomial-time algorithms. Thus, our proposed method not only aligns with the optimal information-theoretic sample limit but also significantly reduces the required sample size compared to other contemporary polynomial-time techniques.

Attaining the optimal sample complexity $\Omega(k \log p)$ in polynomial time requires extra structural assumptions. Indeed, assuming the planted-clique conjecture holds, reductions show that no polynomial-time algorithm can recover the spike with only $\Omega(k \log p)$ samples without additional conditions (Berthet & Rigollet, 2013; Krauthgamer et al., 2015; Wang et al., 2016; Gao et al., 2017; Brennan et al., 2018). We assume $\lambda = \Omega(\|v\|_\infty^{-1})$. In much of the sparse-PCA literature, $\lambda$ is treated as a constant when deriving both information-theoretic lower bounds and algorithmic sample-complexity guarantees. In settings where the nonzeros of $v$ follow a power-law decay– a common model in compressive sensing (Donoho, 2006; Candes et al., 2006)–$\lambda$ naturally remains of constant order under this assumption. An analogous phenomenon appears in sparse phase retrieval (Wang et al., 2017; Cai et al., 2022; 2024), where power-law decay likewise enables optimal $\Omega(k \log p)$ recovery (Jagatap & Hegde, 2019).

Under the assumption $\lambda = \Omega(\|v\|_\infty^{-1})$ from Theorem 3.2, diagonal thresholding (Johnstone & Lu, 2009) cannot attain the optimal $\Omega(k \log p)$ sample complexity. The reason is that diagonal thresholding fails to exploit the decay structure of $v$ in estimation, whereas our method does. By Proposition 3.1, a larger statistical gap implies lower sample complexity. When $\|v\|_\infty$ has constant order, our method's gap in (7) depends only linearly on $\min_{j \in S} |v_j|$, yielding an order-wise improvement over diagonal thresholding's gap in (5); diagonal thresholding cannot exploit the constant-order $\|v\|_\infty$ and thus remains fundamentally limited.

**Theorem 3.3** (Two-stage algorithm). *Let $v \in \mathbb{R}^p$ be a $k$-sparse unit vector. Let $x_i = \sqrt{\lambda} g_i v + \xi_i$, $i = 1, \ldots, n$, where $\lambda > 0$, $g_i \overset{\text{iid}}{\sim} \mathcal{N}(0, 1)$ and $\xi_i \overset{\text{iid}}{\sim} \mathcal{N}(0, I_p)$ are independent. There exist universal constants $C_3, C_4, C_5 > 0$ such that if $\lambda \geq C_3\|v\|_\infty^{-1}$ and $n \geq C_4 k \log p$, then with probability exceeding $1 - 5p^{-1}$, the output $v^t$ of Algorithm 2 with parameter $k' = C_5 k$ and an initial estimate $v^0$ generated by Algorithm 1 satisfies*

$$\text{dist}(v^t, v) \leq \underbrace{d^t \cdot \text{dist}(v, v^0)}_{\text{Optimization error}} + \underbrace{d'\sqrt{k \log p / n}}_{\text{Statistical error}}, \quad (11)$$

*where $0 < d < 1$ and $d' > 0$ are constants.*

The proof of Theorem 3.3 is detailed in Appendix A.4. Theorem 3.3 shows that the estimation error between the estimated and true spikes is bounded by two terms, i.e., the optimization error and statistical error. The optimization error, $d^t \cdot \text{dist}(v, v^0)$, decays to zero at a linear rate with respect to the iteration number $t$. The statistical error of our algorithm is on the order of $\sqrt{\frac{k \log p}{n}}$, achieving the minimax optimal rate (Vu & Lei, 2013; Wang et al., 2014; 2016). Notably, the algorithm attains this optimal rate under the desired sample complexity of $\Omega(k \log p)$, provided $\lambda = \Omega(\|v\|_\infty^{-1})$. We can see the statistical error is independent of $t$, implying that it will not decrease during iterations in the algorithm.

## 4. Experimental Results

We validate our two-stage algorithm in MATLAB on 2.10 GHz Xeon Gold 6152 machines, confirming the theoretical results and demonstrating its efficiency. Estimation error is measured as

$$\text{Error} = \text{dist}(v, \hat{v}),$$

and support recovery by

$$\text{F-score} = \frac{2\text{tp}}{2\text{tp} + \text{fp} + \text{fn}},$$

where tp, fp, and fn denote the counts of true positives, false positives and false negatives; The F-score ranges from 0 to 1, with 1 indicating perfect support recovery.

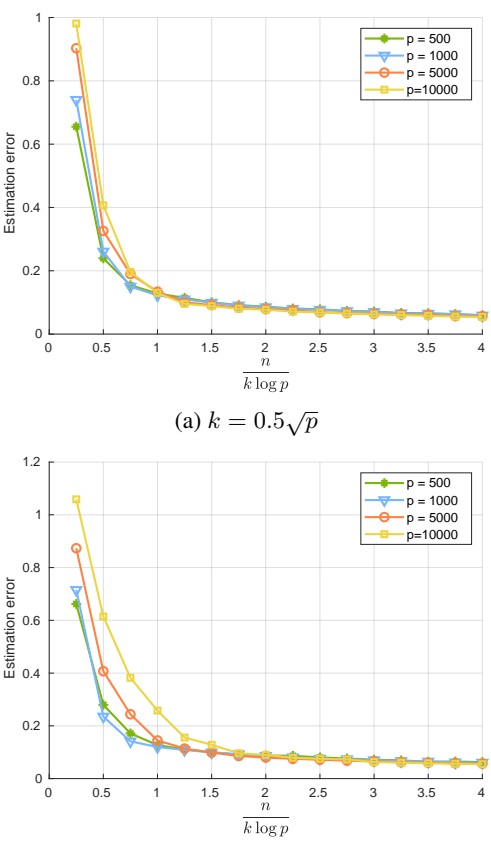

(a) $k = 0.5\sqrt{p}$

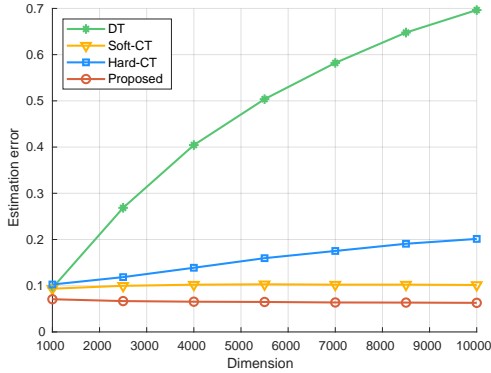

(a) Comparisons of estimation error

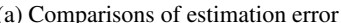

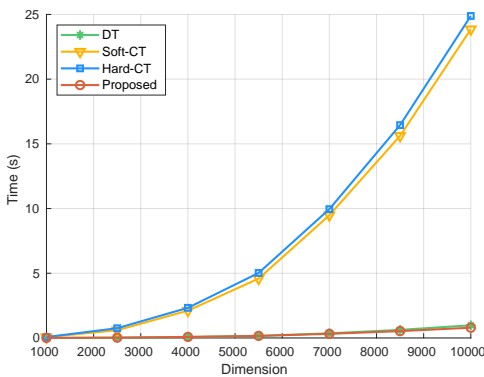

(b) $k = 0.02p$

*Figure 1.* Estimation error versus scaled sample size for our algorithm. We consider two sparsity settings: (a) $k = 0.5\sqrt{p}$ and (b) $k = 0.02p$. Curves are averaged over 200 executions.

(b) Comparisons of computational time

*Figure 2.* Estimation error and computational time versus dimension for different methods. Our algorithm demonstrates both estimation accuracy and computational efficiency.

## 4.1. Validating Theoretical Results

We provide empirical evidence that our algorithm achieves an estimation error of the order $\sqrt{k \log p / n}$, provided the sample size $n \geq \tilde{c} k \log p$ for some constant $\tilde{c}$.

In experiments, we set the dimension $p$ to range from 500 to 10000, with each curve corresponding to a specific dimension. The sparsity $k$ is set to $0.5\sqrt{p}$ in (a) and $0.02p$ in (b). We generate the true spike $v$ with $k$ nonzero entries, where the support of the $k$ entries is chosen randomly, and the value of each entry is either $1/\sqrt{k}$ or $-1/\sqrt{k}$. We then generate $n$ i.i.d. samples $x_i$ from the single-spike model. The signal strength $\lambda$ is set to 10.

Figure 1 plots the estimation error of our algorithm as a function of the scaled sample size $\frac{n}{k \log p}$. When $\frac{n}{k \log p} \geq 2$, the curves corresponding to different dimensions align well with each other, indicating they achieve nearly the same estimation error. For instance, when $\frac{n}{k \log p} = 2$, we have $n = 2k \log p$. Without loss of generality, we consider the estimation error to be $\sqrt{c_{k,p} k \log p / n}$, where $c_{k,p}$ can be a function of $k$ and $p$. By substituting $n = 2k \log p$, the estimation error simplifies to $\sqrt{c_{k,p}/2}$.

Figure 1 shows that the estimation errors are nearly identical for different values of $p$, indicating that $c_{k,p}$ does not depend on $p$. Moreover, since $k$ is a function of $p$ in our settings, $c_{k,p}$ does not depend on $k$ either. Therefore, $c_{k,p}$ is independent of both $p$ and $k$. Consequently, empirical evidence shows that the estimation error of our algorithm is $O(\sqrt{k \log p / n})$, provided the sample size $n \geq 2k \log p$.

## 4.2. Demonstrating Estimation Performance

We demonstrate that our algorithm surpasses state-of-the-art methods in both estimation accuracy and computational efficiency. Diagonal thresholding (DT) (Johnstone & Lu, 2009) tends to produce large estimation errors, whereas covariance thresholding (CT) methods, including soft-thresholding (Soft-CT) (Deshpande & Montanari, 2016) and hard-thresholding (Hard-CT) (Krauthgamer et al., 2015), require substantial computational resources. In contrast, our proposed algorithm consistently achieves superior performance in terms of both lower estimation error and reduced computational time.

Figure 2 compares the estimation error and computational time of different methods across various dimensions. We set the sparsity as $k = 0.02p$ and the sample size $n = 3k \log p$.

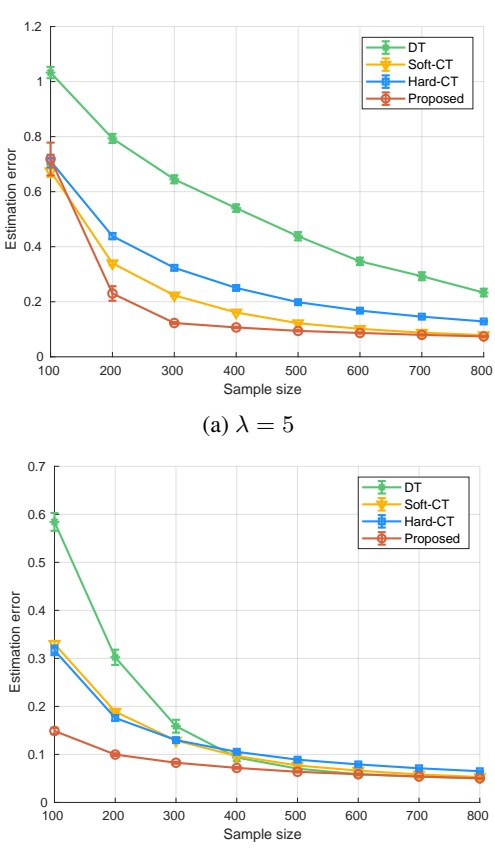

(a) $\lambda = 5$

(b) $\lambda = 10$

*Figure 3.* Estimation error versus sample size for different methods under the settings of (a) $\lambda = 5$ and (b) $\lambda = 10$. The dimension is set to $p = 1000$. Error bars indicate 95% confidence intervals for the mean estimation error, calculated from 200 independent runs.

As the dimension $p$ increases from 1000 to 10000, we observe that the computational time for both Soft-CT and Hard-CT increases dramatically, while DT and our proposed method show a steady increase and take significantly less time. Furthermore, while Soft-CT, Hard-CT, and our algorithm achieve substantially smaller estimation errors compared to DT, our algorithm consistently obtains the smallest estimation error among all methods.

In summary, Figure 2 shows our algorithm excels in both estimation accuracy and computational efficiency. In contrast, the diagonal thresholding method incurs large estimation errors, while the covariance thresholding methods are computationally intensive.

Figure 3 compares the estimation errors of different algorithms as a function of sample size, with the signal strength $\lambda$ set to 5 in (a) and 10 in (b). Our proposed algorithm consistently achieves smaller estimation errors than the other algorithms, except when $n = 100$, where the limited number of samples leads to large estimation errors across all methods. When the signal strength $\lambda$ is increased to 10 in Figure 3(b), similar conclusions can be drawn.

## 4.3. Examining Computational Performance

We assess the scalability of our proposed algorithm by measuring its running time and iteration count as functions of dimension $p$, sample size $n$, and sparsity $k$. In each experiment, we vary one of these parameters while holding the other two fixed to isolate its impact on per-iteration cost and convergence behavior.

Table 1 reveals the algorithm's response to increasing dimension $p$. The results demonstrate that while the per-iteration computational time exhibits near-linear growth with dimension, the iteration count remains remarkably stable. This stability in iteration count is particularly noteworthy, as it results in a total running time that scales approximately linearly with $p$. Such behavior strongly indicates that our method maintains its computational efficiency even in high-dimensional regimes, a crucial advantage for modern large-scale applications.

| Dimension | 5000 | 7500 | 10000 | 12500 | 15000 |
|---|---|---|---|---|---|
| Number of iterations | 9.05 | 9.06 | 9.14 | 9.13 | 9.13 |
| Time for an iteration(s) | 0.018 | 0.029 | 0.040 | 0.048 | 0.055 |
| Total running time(s) | 0.165 | 0.264 | 0.367 | 0.440 | 0.506 |

*Table 1.* Impact of dimension $p$ on computational time in the refinement stage of Algorithm 2, with $n = 2500$ and $k = 100$.

Table 2 presents an interesting interplay between sample size $n$ and computational efficiency. While larger sample sizes induce a linear increase in per-iteration computation time, they simultaneously yield a compensatory decrease in the required number of iterations. This trade-off results in only a modest increase in total running time as $n$ grows. The reduction in iteration count can be attributed to the enhanced statistical accuracy of initial estimates with larger sample sizes—more samples provide a more precise initialization closer to the optimal solution, thereby accelerating convergence.

| Sample size | 2000 | 2500 | 3000 | 3500 | 4000 |
|---|---|---|---|---|---|
| Number of iterations | 9.38 | 8.92 | 8.47 | 7.53 | 5.75 |
| Time for an iteration(s) | 0.014 | 0.019 | 0.022 | 0.026 | 0.030 |
| Total running time(s) | 0.133 | 0.167 | 0.191 | 0.195 | 0.175 |

*Table 2.* Impact of sample size $n$ on computational time in the refinement stage of Algorithm 2, with $p = 5000$ and $k = 100$.

Table 3 demonstrates the algorithm's robustness to varying sparsity levels $k$. The results show that increased sparsity leads to a mild increase in iteration count while maintaining nearly constant per-iteration computation times. This translates to only a slight growth in total running time with increasing $k$, indicating that our method efficiently handles varying degrees of problem sparsity without significant computational overhead.

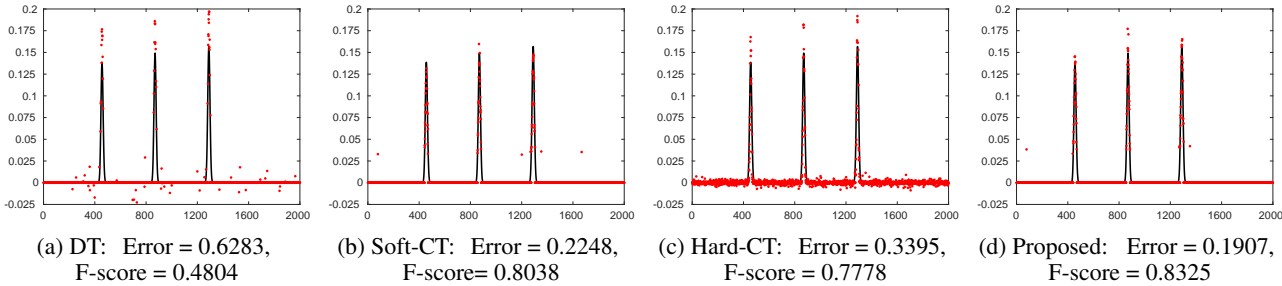

(a) DT: Error = 0.6283, F-score = 0.4804

(b) Soft-CT: Error = 0.2248, F-score= 0.8038

(c) Hard-CT: Error = 0.3395, F-score = 0.7778

(d) Proposed: Error = 0.1907, F-score = 0.8325

*Figure 4.* Comparison of different algorithms in the "three-peak" experiment. The black curve represents the true spike, while the red points indicate the estimated spike by each algorithm. Our algorithm achieves a smaller estimation error and a higher F-score compared to state-of-the-art algorithms, demonstrating superior performance in both estimation accuracy and support recovery.

| Sparsity | 100 | 125 | 150 | 175 | 200 |
|---|---|---|---|---|---|
| Number of iterations | 8.55 | 9.24 | 9.85 | 10.14 | 10.39 |
| Time for an iteration(s) | 0.022 | 0.024 | 0.023 | 0.023 | 0.022 |
| Total running time(s) | 0.189 | 0.219 | 0.224 | 0.234 | 0.231 |

*Table 3.* Impact of sparsity $k$ on computational time in the refinement stage of Algorithm 2, with $p = 5000$ and $n = 3000$.

The empirical results across Tables 1, 2, and 3 provide strong validation of our theoretical complexity analysis. The observed near-linear scaling of per-iteration time with both $p$ and $n$ aligns precisely with the theoretical $O(pn)$ complexity bound. This correspondence between theoretical predictions and empirical measurements, coupled with the favorable iteration scaling properties, establishes our method as both theoretically sound and practically efficient for large-scale sparse principal component analysis.

### 4.4. Experiments on "Three-Peak" Data

The "three-peak" experiment (Johnstone & Lu, 2009) is a widely recognized benchmark for assessing the performance of sparse PCA algorithms. This experiment aims to assess the effectiveness of an algorithm in recovering a structured sparse signal from noisy data. In this setup, the true spike vector $v$ is constructed as a mixture of three Beta densities on $[0, 1]$, producing a characteristic shape with three distinct peaks, as illustrated by the black curve in Figure 4. For our experiments, we set the dimension to $p = 2000$, the sample size to $n = 1000$, and the signal strength to $\lambda = 8$. The choice of these parameters reflect a high-dimensional setting where the number of variables significantly exceeds the number of observations, posing a substantial challenge for accurate signal recovery.

Our proposed algorithm demonstrates superior performance compared to state-of-the-art methods. Specifically, it achieves a smaller estimation error and a higher F-score, reflecting superior accuracy in both estimating the true spike and correctly identifying the support of $v$. These results highlight the advantages of our algorithm in terms of estimation accuracy and support recovery.

## 5. Conclusions

In this paper, we have investigated the single-spiked covariance model for sparse PCA, developing two complementary algorithms. Our thresholding-based algorithm operates in polynomial time and successfully bridges the gap between information-theoretic limits and computational efficiency, requiring only $\Omega(k \log p)$ samples when $\lambda = \Omega(\|v\|_\infty^{-1})$. Building upon this foundation, our two-stage nonconvex algorithm combines the thresholding technique with truncated power iteration to achieve the minimax optimal rate of statistical error under the same sample complexity. Extensive experiments have validated the superior performance of our algorithms in terms of estimation accuracy and computational efficiency.

While this work makes significant progress in addressing the estimation problem for sparse PCA, several promising directions merit further investigation. First, we have not addressed the support recovery problem–specifically, establishing support recovery consistency of the spike with optimal sample complexity–which poses distinct statistical and computational challenges. Second, evaluating the optimality of our signal strength condition and exploring whether this requirement can be relaxed would deepen our understanding of the fundamental limits in sparse PCA.

### Acknowledgement

This work was supported by the Hong Kong Research Grant Council GRFs 16306124, and 16307023, and Hong Kong RGC Postdoctoral Fellowship Scheme of Project No. PDFS2425-6S05. We would also like to thank the anonymous reviewers for their valuable feedback on the manuscript.

### Impact Statement

This paper aims to advance the theoretical and computational aspects of sparse PCA, enabling more efficient high-dimensional data analysis across diverse application domains. We do not foresee any specific ethical concerns arising from this work.

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

# A. Proofs

We present the proofs for Proposition 3.1, Theorem 3.2 for our thresholding algorithm, and Theorem 3.3 for our two-stage algorithm. First, we introduce some technical lemmas in Appendix A.1. Subsequently, we present the proofs for Proposition 3.1, Theorem 3.2, and Theorem 3.3 in Appendices A.2, A.3, and A.4, respectively.

Throughout Appendix A, we define a matrix $\boldsymbol{E} := \hat{\boldsymbol{\Sigma}} - \mathbb{E}\big[\hat{\boldsymbol{\Sigma}}\big]$. For any symmetric matrix $\boldsymbol{A} \in \mathbb{R}^{n \times n}$, we define the largest and smallest $s$-sparse eigenvalue by

$$\lambda_{\max}(\boldsymbol{A}, s) = \max_{\boldsymbol{w} \in \mathbb{R}^n, \|\boldsymbol{w}\|_2 = 1, \|\boldsymbol{w}\|_0 = s} \boldsymbol{w}^T \boldsymbol{A} \boldsymbol{w}, \ \lambda_{\min}(\boldsymbol{A}, s) = \min_{\boldsymbol{w} \in \mathbb{R}^n, \|\boldsymbol{w}\|_2 = 1, \|\boldsymbol{w}\|_0 = s} \boldsymbol{w}^T \boldsymbol{A} \boldsymbol{w},$$

respectively. Also, we define the maximum spectral norm of all $s \times s$ submatrices of $\boldsymbol{A}$ by

$$\rho(\boldsymbol{A}, s) = \max \left\{ |\lambda_{\max}(\boldsymbol{A}, s)|, |\lambda_{\min}(\boldsymbol{A}, s)| \right\}. \tag{12}$$

## A.1. Technical Lemmas

In this section, we introduce and prove some technical lemmas that will be used for the proofs of Theorem 3.2 and Theorem 3.3.

The first lemma bounds the quantity $\rho(\boldsymbol{E}, s)$ defined in (12).

**Lemma A.1.** *For any $t \in (0, 1)$,*

$$\mathbb{P}\left\{ \rho(\boldsymbol{E}, s) \leq 3t\lambda \right\} \geq 1 - 2 \left( \frac{9ep}{s} \right)^s \exp\left( -\frac{C_1' n t^2 \lambda^2}{(\lambda + 1)^2} \right) \tag{13}$$

*for some absolute constant $C_1' > 0$.*

*Proof.* Denote the set of $s$-sparse vectors in $\mathbb{R}^p$ by $\mathbb{T}_s^p := \{ \boldsymbol{w} \mid \|\boldsymbol{w}\|_2 = 1, \|\boldsymbol{w}\|_0 \leq s \}$. For any $\delta \in (0, 1)$, there exists a set $\mathcal{N}_\delta \subset \mathbb{T}_s^p$ such that for any $\boldsymbol{w} \in \mathbb{T}_s^p$, there exists $\boldsymbol{w}_\delta \in \mathcal{N}_\delta$ such that $\operatorname{supp}(\boldsymbol{w}) = \operatorname{supp}(\boldsymbol{w}_\delta)$ and $\|\boldsymbol{w} - \boldsymbol{w}_\delta\|_2 \leq \delta$ and $|\mathcal{N}_\delta| \leq \binom{p}{s} (\frac{3}{\delta})^s \leq (\frac{3ep}{\delta s})^s$ (Baraniuk et al., 2008).

From the definition,

$$\rho(\boldsymbol{E}, s) = \max_{\substack{\boldsymbol{y}, \boldsymbol{z} \in \mathbb{T}_s^p, \\ \operatorname{supp}(\boldsymbol{y}) = \operatorname{supp}(\boldsymbol{z})}} \boldsymbol{y}^T \boldsymbol{E} \boldsymbol{z} =: \boldsymbol{y}_*^T \boldsymbol{E} \boldsymbol{z}_*.$$

From the definition of $\mathcal{N}_\delta$, there exists $\boldsymbol{y}_\delta, \boldsymbol{z}_\delta \in \mathcal{N}_\delta$ such that $\operatorname{supp}(\boldsymbol{y}_\delta) = \operatorname{supp}(\boldsymbol{y}_*) = \operatorname{supp}(\boldsymbol{z}_*) = \operatorname{supp}(\boldsymbol{z}_\delta)$, $\|\boldsymbol{y}_* - \boldsymbol{y}_\delta\|_2 \leq \delta$ and $\|\boldsymbol{z}_* - \boldsymbol{z}_\delta\|_2 \leq \delta$. Then we have

$$\begin{aligned}
\boldsymbol{y}_*^T \boldsymbol{E} \boldsymbol{z}_* &= \boldsymbol{y}_*^T \boldsymbol{E} (\boldsymbol{z}_* - \boldsymbol{z}_\delta) + (\boldsymbol{y}_* - \boldsymbol{y}_\delta)^T \boldsymbol{E} \boldsymbol{z}_\delta + \boldsymbol{y}_\delta^T \boldsymbol{E} \boldsymbol{z}_\delta \\
&\leq 2\delta \boldsymbol{y}_*^T \boldsymbol{E} \boldsymbol{z}_* + \boldsymbol{y}_\delta^T \boldsymbol{E} \boldsymbol{z}_\delta,
\end{aligned}$$

which implies that

$$\rho(\boldsymbol{E}, s) \leq (1 - 2\delta)^{-1} \boldsymbol{y}_\delta^T \boldsymbol{E} \boldsymbol{z}_\delta \leq (1 - 2\delta)^{-1} \max_{\substack{\boldsymbol{y}, \boldsymbol{z} \in \mathcal{N}_\delta, \\ \operatorname{supp}(\boldsymbol{y}) = \operatorname{supp}(\boldsymbol{z})}} \boldsymbol{y}^T \boldsymbol{E} \boldsymbol{z}, \tag{14}$$

where the inequalities hold when $1 - 2\delta > 0$.

Now for any $(\boldsymbol{y}, \boldsymbol{z}) \in \mathcal{N}_\delta$, we bound $\left| \boldsymbol{y}^T \boldsymbol{E} \boldsymbol{z} \right|$ as follows. Recall that

$$\boldsymbol{y}^T \boldsymbol{E} \boldsymbol{z} = \frac{1}{n} \sum_{i=1}^n \underbrace{\boldsymbol{y}^T \left( \boldsymbol{x}_i \boldsymbol{x}_i^T - \mathbb{E}\big[ \left[ \boldsymbol{x}_i \boldsymbol{x}_i^T \right] \big] \right) \boldsymbol{z}}_{=: \boldsymbol{Z}_i},$$

where $\boldsymbol{x}_i = \sqrt{\lambda} g_i \boldsymbol{v} + \boldsymbol{\xi}_i$. With the assumption of $g_i$ and $\boldsymbol{\xi}_i$, we have $\boldsymbol{y}^T \boldsymbol{x}_i \overset{\text{iid}}{\sim} \mathcal{N}(0, \lambda(\boldsymbol{y}^T \boldsymbol{v})^2 + 1)$ and $\boldsymbol{x}_i^T \boldsymbol{z} \overset{\text{iid}}{\sim} \mathcal{N}(0, \lambda(\boldsymbol{z}^T \boldsymbol{v})^2 + 1)$, hence $\boldsymbol{y}^T \boldsymbol{x}_i \boldsymbol{x}_i^T \boldsymbol{z}$ is sub-exponential (Vershynin, 2018). Using the Centering Lemma for sub-exponential random variables (Vershynin, 2018), we have

$$\|\boldsymbol{Z}_i\|_{\Psi_1} \leq c_1 \sqrt{\lambda(\boldsymbol{y}^T \boldsymbol{v})^2 + 1} \sqrt{\lambda(\boldsymbol{z}^T \boldsymbol{v})^2 + 1} \leq c_1(\lambda + 1),$$

where $c_1 > 0$ is some absolute constant and the last inequality holds since $\|\boldsymbol{y}\|_2 = \|\boldsymbol{z}\|_2 = \|\boldsymbol{v}\|_2 = 1$. Therefore, by Bernstein's inequality (Vershynin, 2018), for any $t \in (0, 1)$, it holds that

$$\mathbb{P}\left\{\left|\boldsymbol{y}^T \boldsymbol{E} \boldsymbol{z}\right| \geq t\lambda\right\} \leq 2\exp\left[-C_1' n \frac{t^2\lambda^2}{(\lambda+1)^2}\right], \tag{15}$$

where $C_1' > 0$ is some absolute constant. By taking union bounds for all $(\boldsymbol{y}, \boldsymbol{z}) \in \mathcal{N}_\delta$, we obtain that, with probability exceeding $1 - 2(\frac{3ep}{\delta s})^s \exp[-C_1' n \frac{t^2\lambda^2}{(\lambda+1)^2}]$,

$$\max_{\substack{\boldsymbol{y},\boldsymbol{z}\in\mathcal{N}_\delta, \\ \operatorname{supp}(\boldsymbol{y})=\operatorname{supp}(\boldsymbol{z})}} \left|\boldsymbol{y}^T \boldsymbol{E} \boldsymbol{z}\right| \leq t\lambda.$$

Setting $\delta = \frac{1}{3}$ together with (14) leads to (13). $\qquad\square$

The next lemma gives the concentration inequality for $\chi^2$-square variables.

**Lemma A.2** ((Laurent & Massart, 2000))**.** $\chi^2_{(m)}$ *denotes a central chi-squared variable with $m$ degrees of freedom. For all $t \geq 0$,*

$$\mathbb{P}[\chi^2_{(m)} - m \geq 2\sqrt{mt} + 2t] \leq \exp(-t), \tag{16}$$

$$\mathbb{P}[\chi^2_{(m)} - m \leq -2\sqrt{mt}] \leq \exp(-t). \tag{17}$$

The next lemma bounds the error between $\boldsymbol{v}$ and a $s$-sparse largest eigenvector of $\hat{\boldsymbol{\Sigma}}$ in unit length.

**Lemma A.3.** *Let $\Lambda \subset [p]$ be such that $\Lambda \bigcup \mathcal{S} \neq \emptyset$ and $|\Lambda| = s$. Let $\boldsymbol{u}$ be an eigenvector of unit length corresponding to the largest eigenvalue of $\hat{\boldsymbol{\Sigma}}_\Lambda$. If $\rho(\boldsymbol{E}, s) < \frac{\lambda}{2}\|\boldsymbol{v}_\Lambda\|_2^2$, then we have*

$$\operatorname{dist}(\boldsymbol{u}, \boldsymbol{v}_\Lambda)^2 \leq \|\boldsymbol{v}_\Lambda\|_2^2 + 1 - 2\frac{\|\boldsymbol{v}_\Lambda\|_2}{\sqrt{1 + \frac{\rho^2(\boldsymbol{E},s)}{\left(\lambda\|\boldsymbol{v}_\Lambda\|_2^2 - 2\rho(\boldsymbol{E},s)\right)^2}}}.$$

*Proof.* Recall that $\boldsymbol{E} = \hat{\boldsymbol{\Sigma}} - \mathbb{E}[\hat{\boldsymbol{\Sigma}}]$. Denote $\bar{\lambda}$ the largest eigenvalue of $\hat{\boldsymbol{\Sigma}}_\Lambda$, i.e. $\bar{\lambda} = \lambda_1(\hat{\boldsymbol{\Sigma}}_\Lambda)$. From Weyl's inequality (Horn & Johnson, 2012), we have

$$\bar{\lambda} \geq \lambda_1(\mathbb{E}[\hat{\boldsymbol{\Sigma}}_\Lambda]) + \lambda_n(\boldsymbol{E}_\Lambda) \geq \lambda\|\boldsymbol{v}_\Lambda\|_2^2 + 1 - \rho(\boldsymbol{E}, s), \tag{18}$$

where the last inequality holds since $\mathbb{E}[\hat{\boldsymbol{\Sigma}}] = \lambda \boldsymbol{v}\boldsymbol{v}^T + \boldsymbol{I}$ and $\lambda_n(\boldsymbol{E}_\Lambda) \geq -\rho(\boldsymbol{E}, s)$ from the definition. Similarly, we have for all $j \geq 2$,

$$\left|\lambda_j(\hat{\boldsymbol{\Sigma}}_\Lambda)\right| \leq \left|\lambda_j(\mathbb{E}[\hat{\boldsymbol{\Sigma}}_\Lambda])\right| + \left|\lambda_j(\hat{\boldsymbol{\Sigma}}_\Lambda) - \lambda_j(\mathbb{E}[\hat{\boldsymbol{\Sigma}}_\Lambda])\right| = 1 + \max\left\{|\lambda_1(\boldsymbol{E}_\Lambda)|, |\lambda_n(\boldsymbol{E}_\Lambda)|\right\} \leq 1 + \rho(\boldsymbol{E}, s). \tag{19}$$

Notice $\|\boldsymbol{u}\|_2 = 1$ but $\|\boldsymbol{v}_\Lambda\|_2 \leq 1$. We divide $\boldsymbol{u}$ as

$$\boldsymbol{u} = a_1 \frac{\boldsymbol{v}_\Lambda}{\|\boldsymbol{v}_\Lambda\|_2} + a_2 \boldsymbol{y}$$

with $\boldsymbol{v}_\Lambda^T \boldsymbol{y} = 0$, $\|\boldsymbol{y}\|_2 = 1$ and $a_1^2 + a_2^2 = 1$. Then we have $\operatorname{supp}(\boldsymbol{y}) \subset \Lambda$, and

$$\bar{\lambda} a_1 \frac{\boldsymbol{v}_\Lambda}{\|\boldsymbol{v}_\Lambda\|_2} + \bar{\lambda} a_2 \boldsymbol{y} = \bar{\lambda} \boldsymbol{u} = \hat{\boldsymbol{\Sigma}}_\Lambda \boldsymbol{u} = a_1 \frac{\hat{\boldsymbol{\Sigma}}_\Lambda \boldsymbol{v}_\Lambda}{\|\boldsymbol{v}_\Lambda\|_2} + a_2 \hat{\boldsymbol{\Sigma}}_\Lambda \boldsymbol{y}.$$

By taking the inner product with $\boldsymbol{y}$, we obtain

$$\bar{\lambda} a_2 = a_1 \frac{\boldsymbol{y}^T \hat{\boldsymbol{\Sigma}}_\Lambda \boldsymbol{v}_\Lambda}{\|\boldsymbol{v}_\Lambda\|_2} + a_2 \boldsymbol{y}^T \hat{\boldsymbol{\Sigma}}_\Lambda \boldsymbol{y}.$$

Since $\boldsymbol{v}_\Lambda$ is the eigenvector of $\mathbb{E}\big[\hat{\boldsymbol{\Sigma}}_\Lambda\big]$ and $\boldsymbol{v}_\Lambda^T \boldsymbol{y} = 0$, we have $\boldsymbol{y}^T \mathbb{E}\big[\hat{\boldsymbol{\Sigma}}_\Lambda\big]\boldsymbol{v}_\Lambda = 0$. This leads to

$$|a_2| = |a_1| \frac{\left| \boldsymbol{y}^T (\boldsymbol{E}_\Lambda + \mathbb{E}\big[\hat{\boldsymbol{\Sigma}}_\Lambda\big]) \frac{\boldsymbol{v}_\Lambda}{\|\boldsymbol{v}_\Lambda\|_2} \right|}{\left| \bar{\lambda} - \boldsymbol{y}^T \hat{\boldsymbol{\Sigma}}_\Lambda \boldsymbol{y} \right|} = |a_1| \frac{\left| \boldsymbol{y}^T \boldsymbol{E}_\Lambda \frac{\boldsymbol{v}_\Lambda}{\|\boldsymbol{v}_\Lambda\|_2} \right|}{\left| \bar{\lambda} - \boldsymbol{y}^T \hat{\boldsymbol{\Sigma}}_\Lambda \boldsymbol{y} \right|}.$$

Since $\mathrm{supp}(\boldsymbol{y}) \subset \Lambda$, we have $\left| \boldsymbol{y}^T \boldsymbol{E}_\Lambda \frac{\boldsymbol{v}_\Lambda}{\|\boldsymbol{v}_\Lambda\|_2} \right| \leq \rho(\boldsymbol{E}, s)$. Moreover, since $\boldsymbol{y}$ is perpendicular to $\boldsymbol{v}_\Lambda$, from (19) we have

$$\left| \boldsymbol{y}^T \hat{\boldsymbol{\Sigma}}_\Lambda \boldsymbol{y} \right| \leq \max_{j \geq 2} \left| \lambda_j(\hat{\boldsymbol{\Sigma}}_\Lambda) \right| \leq 1 + \rho(\boldsymbol{E}, s).$$

So from (18) and $\rho(\boldsymbol{E}, s) < \frac{\lambda}{2}\|\boldsymbol{v}_\Lambda\|_2^2$, we have

$$\frac{|a_2|}{|a_1|} = \frac{\left| \boldsymbol{y}^T \boldsymbol{E}_\Lambda \frac{\boldsymbol{v}_\Lambda}{\|\boldsymbol{v}_\Lambda\|_2} \right|}{\left| \bar{\lambda} - \boldsymbol{y}^T \hat{\boldsymbol{\Sigma}}_\Lambda \boldsymbol{y} \right|} \leq \frac{\rho(\boldsymbol{E}, s)}{\lambda\|\boldsymbol{v}_\Lambda\|_2^2 - 2\rho(\boldsymbol{E}, s)}.$$

Then, since $a_1^2 + a_2^2 = 1$, we have

$$a_1^2 \geq \frac{1}{1 + \frac{\rho^2(\boldsymbol{E}, s)}{\left( \lambda\|\boldsymbol{v}_\Lambda\|_2^2 - 2\rho(\boldsymbol{E}, s) \right)^2}},$$

which implies that

$$\begin{aligned}
\mathrm{dist}(\boldsymbol{u}, \boldsymbol{v}_\Lambda)^2 &= \min\left\{ \|\boldsymbol{v}_\Lambda - \boldsymbol{u}\|_2^2, \|\boldsymbol{v}_\Lambda + \boldsymbol{u}\|_2^2 \right\} \\
&= \|\boldsymbol{v}_\Lambda\|_2^2 + 1 - 2\,|a_1| \cdot \|\boldsymbol{v}_\Lambda\|_2 \\
&\leq \|\boldsymbol{v}_\Lambda\|_2^2 + 1 - 2\frac{\|\boldsymbol{v}_\Lambda\|_2}{\sqrt{1 + \frac{\rho^2(\boldsymbol{E}, s)}{\left( \lambda\|\boldsymbol{v}_\Lambda\|_2^2 - 2\rho(\boldsymbol{E}, s) \right)^2}}}.
\end{aligned}$$

$\square$

The following two lemmas are used to prove the convergence of truncated power method in our proposed two-stage algorithm.

**Lemma A.4** ((Yuan & Zhang, 2013))**.** *Let $\boldsymbol{z}$ be the eigenvector with the largest eigenvalue (in absolute value) of a symmetric matrix $\boldsymbol{A}$, and let $\kappa < 1$ be the ratio of the second to the largest eigenvalue in absolute values. Given any $\boldsymbol{y}$ such that $\|\boldsymbol{y}\|_2 = 1$, let $\boldsymbol{y}' = \boldsymbol{A}\boldsymbol{y}/\|\boldsymbol{A}\boldsymbol{y}\|_2$, then*

$$\left| \boldsymbol{z}^T \boldsymbol{y}' \right| \geq \left| \boldsymbol{z}^T \boldsymbol{y} \right| \left( 1 + \frac{1}{2}\left( 1 - \kappa^2 \right)\left( 1 - \left| \boldsymbol{z}^T \boldsymbol{y} \right|^2 \right) \right).$$

**Lemma A.5** ((Yuan & Zhang, 2013))**.** *Consider $\boldsymbol{y}$ with $\|\boldsymbol{y}\|_0 = s$. Consider $\boldsymbol{z}$ and let $\mathcal{F} = \mathrm{supp}(\boldsymbol{z}, s')$ be the $s'$ indices with the largest absolute values in $\boldsymbol{z}$. If $\|\boldsymbol{y}\|_2 = \|\boldsymbol{z}\|_2 = 1$, then*

$$\left| \boldsymbol{y}^T \boldsymbol{z}_\mathcal{F} \right| \geq \left| \boldsymbol{y}^T \boldsymbol{z} \right| - \sqrt{s/s'} \min\left\{ \sqrt{1 - |\boldsymbol{y}^T \boldsymbol{z}|^2}, \left( 1 + \sqrt{s/s'} \right)\left( 1 - \left| \boldsymbol{y}^T \boldsymbol{z} \right|^2 \right) \right\}.$$

### A.2. Proof of Proposition 3.1

*Proof of Proposition 3.1.* For any $j \in \mathcal{S}$ and any $j' \in \mathcal{S}^c$, using (4) and (5) obtains:

$$\left| \hat{\boldsymbol{\Sigma}}_{jj} \right| \geq \left| \big[ \mathbb{E}[\hat{\boldsymbol{\Sigma}}] \big]_{jj} \right| - \left| \hat{\boldsymbol{\Sigma}}_{jj} - \big[ \mathbb{E}[\hat{\boldsymbol{\Sigma}}] \big]_{jj} \right| \geq \lambda v_j^2 + 1 - \frac{1}{2}g_\mathrm{d},$$

$$\left| \hat{\boldsymbol{\Sigma}}_{j'j'} \right| \leq \left| \big[ \mathbb{E}[\hat{\boldsymbol{\Sigma}}] \big]_{j'j'} \right| + \left| \big[ \mathbb{E}[\hat{\boldsymbol{\Sigma}}] \big]_{j'j'} - \hat{\boldsymbol{\Sigma}}_{j'j'} \right| \leq 1 + \frac{1}{2}g_\mathrm{d}.$$

Following from the fact that $g_\mathrm{d} = \lambda \cdot \min_{j \in \mathcal{S}} v_j^2$, one has $\left| \hat{\boldsymbol{\Sigma}}_{jj} \right| \geq \left| \hat{\boldsymbol{\Sigma}}_{j'j'} \right|$.

$\square$

## A.3. Proof of Theorem 3.2

We organize the proof of Theorem 3.2 in three steps. First, we prove that the index $j_0$ chosen in Algorithm 1 satisfies $|v_{j_0}| \geq \frac{\|v\|_\infty}{2}$ with high probability. Second, we show that $\hat{S}$ chosen in Algorithm 1 contains the indices of most of larger non-zero entries of $v$. Finally, we put everything together.

*Step 1: Estimating $|v_{j_0}|$.* Recall that $j_0 = \arg \max\limits_{1 \leq j \leq n} \hat{\Sigma}_{j,j}$.

**Lemma A.6.** *Assume that $\lambda \geq C_2' \|v\|_\infty^{-1}$ and $n \geq C_3' k \log p$ for some absolute constants $C_2', C_3' > 0$. Then, with probability exceeding $1 - p^{-1}$, $|v_{j_0}| \geq \frac{\|v\|_\infty}{2}$.*

*Proof.* Recall that

$$\hat{\Sigma}_{j,j} = \frac{1}{n} \sum_{i=1}^n (\sqrt{\lambda} g_i v_j + \xi_{i,j})(\sqrt{\lambda} g_i v_j + \xi_{i,j}).$$

Then we have $\mathbb{E}[\hat{\Sigma}_{j,j}] = \lambda v_j^2 + 1$. Since $\sqrt{\lambda} g_i v_j + \xi_{i,j} \overset{\text{iid}}{\sim} \mathcal{N}(0, \lambda v_j^2 + 1)$, we obtain $n \hat{\Sigma}_{j,j}/(\lambda v_j^2 + 1) \overset{\text{iid}}{\sim} \chi_{(n)}^2$.

Firstly, we consider $\hat{\Sigma}_{j_*, j_*}$, where $j_*$ satisfies $|v_{j_*}| = \|v\|_\infty$. Then (17) in Lemma A.2 implies that for any $\varepsilon > 0$,

$$\mathbb{P}\left\{\hat{\Sigma}_{j_*, j_*} \leq (1-\varepsilon)\lambda\|v\|_\infty^2 + 1\right\} = \mathbb{P}\left\{\chi_{(n)}^2 - n \leq -n\frac{\varepsilon\lambda\|v\|_\infty^2}{\lambda\|v\|_\infty^2 + 1}\right\}$$
$$\leq \exp[-\frac{n}{4}\frac{\varepsilon^2\lambda^2\|v\|_\infty^4}{(\lambda\|v\|_\infty^2 + 1)^2}]. \tag{20}$$

Secondly, we consider $\mathcal{S}_1 := \left\{j \in [p] \mid |v_j| < \frac{\|v\|_\infty}{2}\right\}$. By using (16) in Lemma A.2 and taking union bound, we have,

$$\mathbb{P}\left\{\max_{j \in \mathcal{S}_1} \hat{\Sigma}_{j,j} \geq (1-\varepsilon)\lambda\|v\|_\infty^2 + 1\right\}$$
$$\leq (p-1)\mathbb{P}\left\{\chi_{(n)}^2 \geq n\frac{(1-\varepsilon)\lambda\|v\|_\infty^2 + 1}{\frac{1}{4}\lambda\|v\|_\infty^2 + 1}\right\}$$
$$\leq (p-1)\exp[-\frac{n\frac{(\frac{3}{4}-\varepsilon)^2\lambda^2\|v\|_\infty^4}{(\frac{1}{4}\lambda\|v\|_\infty^2+1)^2}}{(1 + \sqrt{\frac{(\frac{7}{4}-2\varepsilon)\lambda\|v\|_\infty^2 + 1}{\frac{1}{4}\lambda\|v\|_\infty^2 + 1}})^2}] \tag{21}$$
$$\leq (p-1)\exp[-\frac{(\frac{3}{4}-\varepsilon)^2 n\lambda^2\|v\|_\infty^4}{4(\frac{1}{4}\lambda\|v\|_\infty^2 + 1)(1 + (1-\varepsilon)\lambda\|v\|_\infty^2)}],$$

where the second inequality holds from the definition of $\mathcal{S}_1$.

Now we combine (20)(21) and set $\varepsilon = \frac{1}{2}$. The complementary events in (20)(21) lead to $\max_{j \in \mathcal{S}_1} \hat{\Sigma}_{j,j} < \frac{1}{2}\lambda\|v\|_\infty^2 + 1 < \hat{\Sigma}_{j_*, j_*} \leq \hat{\Sigma}_{j_0, j_0}$, which implies that $j_0 \notin \mathcal{S}_1$, i.e. $|v_{j_0}| \geq \frac{\|v\|_\infty}{2}$. Therefore, we obtain

$$\mathbb{P}\left\{|v_{j_0}| \geq \frac{\|v\|_\infty}{2}\right\} \geq 1 - p\exp[-\frac{n\lambda^2\|v\|_\infty^4}{16(\lambda\|v\|_\infty^2 + 1)^2}]. \tag{22}$$

Together with the assumptions of $\lambda$ and $n$, (22) leads to the desired result. $\square$

*Remark* A.7. From (22), a sufficient condition for constants $C_2'$ and $C_3'$ in Lemma A.6 is

$$\sqrt{C_3'}\frac{C_2'}{C_2'\|v\|_\infty + 1} \geq \frac{4\sqrt{2}}{\sqrt{k}\|v\|_\infty},$$

which can be simplified as

$$\sqrt{C_3'}\frac{C_2'}{C_2' + 1} \geq 4\sqrt{2}.$$

*Step 2: Estimating $\|\boldsymbol{v}_{\hat{S}}\|_2$.* For any $\gamma \in (0, 1]$, we define $\mathcal{S}_\gamma^- := \left\{ j \in \mathcal{S} \mid |v_j| < \frac{\gamma}{2\sqrt{k}} \right\}$ and $\mathcal{S}_\gamma^+ = \mathcal{S} \setminus \mathcal{S}_\gamma^-$. Then we have $\|\boldsymbol{v}_{\mathcal{S}_\gamma^-}\|_2^2 < \frac{\gamma^2}{4k} \cdot k = \frac{\gamma^2}{4}$ and $\|\boldsymbol{v}_{\mathcal{S}_\gamma^+}\|_2^2 \geq 1 - \frac{\gamma^2}{4}$. Since $\|\boldsymbol{v}\|_\infty \geq \frac{1}{\sqrt{k}}$, Lemma A.6 implies that $|v_{j_0}| \geq \frac{1}{2\sqrt{k}} \geq \frac{\gamma}{2\sqrt{k}}$ with high probability, and thus $j_0 \in \mathcal{S}_\gamma^+$. The following lemma shows that $\mathcal{S}_\gamma^+ \subset \hat{\mathcal{S}}$ with high probability, where $\hat{\mathcal{S}}$ is chosen in Algorithm 1.

**Lemma A.8.** *For any $\gamma \in (0, 1]$, if $\lambda \geq C_4' \|\boldsymbol{v}\|_\infty^{-1}$ and $n \geq C_5' \gamma^{-2} k \log p$ for some absolute constants $C_4', C_5' > 0$, then, with probability exceeding $1 - 3p^{-1}$, $\mathcal{S}_\gamma^+ \subset \hat{\mathcal{S}}$.*

*Proof.* It suffices to show that with high probability,

$$\min_{j \in \mathcal{S}_\gamma^+} \left| \hat{\boldsymbol{\Sigma}}_{j, j_0} \right| > \max_{j \in \mathcal{S}^c} \left| \hat{\boldsymbol{\Sigma}}_{j, j_0} \right|.$$

To prove this, first, we show that for any $l \in \mathcal{S}_2$, where $\mathcal{S}_2 := \left\{ j \in \mathcal{S} \mid |v_j| \geq \frac{\|\boldsymbol{v}\|_\infty}{2} \right\}$,

$$\min_{j \in \mathcal{S}_\gamma^+} \left| \hat{\boldsymbol{\Sigma}}_{j, l} \right| > \max_{j \in \mathcal{S}^c} \left| \hat{\boldsymbol{\Sigma}}_{j, l} \right|,$$

which needs to bound $\left| \hat{\boldsymbol{\Sigma}}_{j, l} \right|$ and $\left| \hat{\boldsymbol{\Sigma}}_{j, l} - \mathbb{E}\left[ \hat{\boldsymbol{\Sigma}}_{j, l} \right] \right|$ for all $j \in \mathcal{S}^c$ and $j \in \mathcal{S}_\gamma^+$.

For any $l \in \mathcal{S}_2$, we consider $\max_{j \in \mathcal{S}^c} \left| \hat{\boldsymbol{\Sigma}}_{j, l} \right|$. From the definition, for all $j \in \mathcal{S}^c$,

$$\hat{\boldsymbol{\Sigma}}_{j, l} = \frac{1}{n} \sum_{i=1}^n \boldsymbol{\xi}_{i, j} (\sqrt{\lambda} g_i v_l + \boldsymbol{\xi}_{i, l}),$$

hence we have $\mathbb{E}\left[ \hat{\boldsymbol{\Sigma}}_{j, l} \right] = 0$. Since $\boldsymbol{\xi}_{i, j} \overset{\text{iid}}{\sim} \mathcal{N}(0, 1)$ and $\sqrt{\lambda} g_i v_l + \boldsymbol{\xi}_{i, l} \overset{\text{iid}}{\sim} \mathcal{N}(0, \lambda v_l^2 + 1)$, $\boldsymbol{\xi}_{i, j}(\sqrt{\lambda} g_i v_l + \boldsymbol{\xi}_{i, l})$ is sub-exponential (Vershynin, 2018) with

$$\|\boldsymbol{\xi}_{i, j}(\sqrt{\lambda} g_i v_l + \boldsymbol{\xi}_{i, l})\|_{\Psi_1} \leq c_3 \sqrt{\lambda \|\boldsymbol{v}\|_\infty^2 + 1},$$

where $c_3 > 0$ is some absolute constant. Then, by taking union bound and using Bernstein's inequality (Vershynin, 2018), for any $\varepsilon_1 > 0$, we have

$$\mathbb{P}\left\{ \max_{j \in \mathcal{S}^c} \left| \hat{\boldsymbol{\Sigma}}_{j, l} \right| \geq \varepsilon_1 \right\} \leq 2(p - k) \exp\left[ -c_4 n \min\left\{ \frac{\varepsilon_1^2}{\lambda \|\boldsymbol{v}\|_\infty^2 + 1}, \frac{\varepsilon_1}{\sqrt{\lambda \|\boldsymbol{v}\|_\infty^2 + 1}} \right\} \right], \tag{23}$$

where $c_4 > 0$ is some absolute constant.

For any $l \in \mathcal{S}_2$, we consider $\max_{j \in \mathcal{S}_\gamma^+} \left| \hat{\boldsymbol{\Sigma}}_{j, j_0} - \mathbb{E}\left[ \hat{\boldsymbol{\Sigma}}_{j, j_0} \right] \right|$. From the definition, for all $j \in \mathcal{S}_\gamma^+$,

$$\hat{\boldsymbol{\Sigma}}_{j, l} = \frac{1}{n} \sum_{i=1}^n (\sqrt{\lambda} g_i v_j + \boldsymbol{\xi}_{i, j})(\sqrt{\lambda} g_i v_l + \boldsymbol{\xi}_{i, l}),$$

which implies that $\mathbb{E}\left[ \hat{\boldsymbol{\Sigma}}_{j, l} \right] \geq \lambda v_j v_l$ for all $j \in \mathcal{S}_\gamma^+$, and thus $\left| \mathbb{E}\left[ \hat{\boldsymbol{\Sigma}}_{j, l} \right] \right| \geq \frac{\gamma \lambda \|\boldsymbol{v}\|_\infty}{4\sqrt{k}}$. On the other hand, since $\sqrt{\lambda} g_i v_j + \boldsymbol{\xi}_{i, j} \overset{\text{iid}}{\sim} \mathcal{N}(0, \lambda v_j^2 + 1)$, $(\sqrt{\lambda} g_i v_j + \boldsymbol{\xi}_{i, j})(\sqrt{\lambda} g_i v_l + \boldsymbol{\xi}_{i, l})$ is sub-exponential (Vershynin, 2018). Using the Centering Lemma for sub-exponential random variables (Vershynin, 2018), we have

$$\|(\sqrt{\lambda} g_i v_j + \boldsymbol{\xi}_{i, j})(\sqrt{\lambda} g_i v_l + \boldsymbol{\xi}_{i, l}) - \mathbb{E}\left[ \hat{\boldsymbol{\Sigma}}_{j, l} \right]\|_{\Psi_1} \leq c_5 (\lambda \|\boldsymbol{v}\|_\infty^2 + 1),$$

where $c_5 > 0$ is an absolute constant. Then, from Bernstein's inequality (Vershynin, 2018), we have

$$\mathbb{P}\left\{ \max_{j \in \mathcal{S}_\gamma^+} \left| \hat{\boldsymbol{\Sigma}}_{j, l} - \mathbb{E}\left[ \hat{\boldsymbol{\Sigma}}_{j, l} \right] \right| \geq \varepsilon_2 \right\} \leq 2k \exp\left[ -c_6 n \min\left\{ \frac{\varepsilon_2^2}{(\lambda \|\boldsymbol{v}\|_\infty^2 + 1)^2}, \frac{\varepsilon_2}{\lambda \|\boldsymbol{v}\|_\infty^2 + 1} \right\} \right], \tag{24}$$

where $c_6 > 0$ is an absolute constant.

Now we combine (23)(24) and set $\varepsilon_1 = \varepsilon_2 = \frac{\gamma\lambda\|\boldsymbol{v}\|_\infty}{8\sqrt{k}}$. The complementary event in (23) is

$$\max_{j\in\mathcal{S}^c}\left|\hat{\boldsymbol{\Sigma}}_{j,l}\right| \le \frac{\gamma\lambda\|\boldsymbol{v}\|_\infty}{8\sqrt{k}}.$$

Moreover, the complementary event in (24) leads to

$$\left|\hat{\boldsymbol{\Sigma}}_{j,l}\right| > \left|\left|\mathbb{E}\left[\hat{\boldsymbol{\Sigma}}_{j,l}\right]\right| - \left|\hat{\boldsymbol{\Sigma}}_{j,l} - \mathbb{E}\left[\hat{\boldsymbol{\Sigma}}_{j,j_0}\right]\right|\right| > \frac{\gamma\lambda\|\boldsymbol{v}\|_\infty}{4\sqrt{k}} - \frac{\gamma\lambda\|\boldsymbol{v}\|_\infty}{8\sqrt{k}} = \frac{\gamma\lambda\|\boldsymbol{v}\|_\infty}{8\sqrt{k}}, \forall j\in\mathcal{S}_\gamma^+.$$

These two inequalities implies that $\min\limits_{j\in\mathcal{S}_\gamma^+}\left|\hat{\boldsymbol{\Sigma}}_{j,l}\right| > \max\limits_{j\in\mathcal{S}^c}\left|\hat{\boldsymbol{\Sigma}}_{j,l}\right|$ for any $l\in\mathcal{S}_2$.

Finally, by taking union bound and using (22)(23)(24),

$$\begin{aligned}
\mathbb{P}\left\{\mathcal{S}_\gamma^+ \subset \hat{\mathcal{S}}\right\} &= \sum_{l\in\mathcal{S}_2}\mathbb{P}\left\{\min_{j\in\mathcal{S}_\gamma^+}\left|\hat{\boldsymbol{\Sigma}}_{j,l}\right| > \max_{j\in\mathcal{S}^c}\left|\hat{\boldsymbol{\Sigma}}_{j,l}\right|, j_0 = l\right\} \\
&\ge \sum_{l\in\mathcal{S}_2}(1 - \mathbb{P}\left\{\min_{j\in\mathcal{S}_\gamma^+}\left|\hat{\boldsymbol{\Sigma}}_{j,l}\right| \le \max_{j\in\mathcal{S}^c}\left|\hat{\boldsymbol{\Sigma}}_{j,l}\right|\right\} - \mathbb{P}\left\{j_0\ne l\right\}) \\
&\ge \sum_{l\in\mathcal{S}_2}(\mathbb{P}\left\{j_0 = l\right\} - \mathbb{P}\left\{\min_{j\in\mathcal{S}_\gamma^+}\left|\hat{\boldsymbol{\Sigma}}_{j,l}\right| \le \max_{j\in\mathcal{S}^c}\left|\hat{\boldsymbol{\Sigma}}_{j,l}\right|\right\}) \\
&\ge 1 - p\exp[-\frac{n\lambda^2\|\boldsymbol{v}\|_\infty^4}{16(\lambda\|\boldsymbol{v}\|_\infty^2 + 1)^2}] - \\
&\quad 2p^2\exp[-c_7 n\min\left\{\frac{\gamma^2\lambda^2\|\boldsymbol{v}\|_\infty^2}{64k(\lambda\|\boldsymbol{v}\|_\infty^2 + 1)^2}, \frac{\gamma\lambda\|\boldsymbol{v}\|_\infty}{8\sqrt{k}(\lambda\|\boldsymbol{v}\|_\infty^2 + 1)}\right\}].
\end{aligned} \tag{25}$$

where $c_7 = \min\{c_4, c_6\}$. Together with the assumption of $\lambda$ and $n$ and since $\gamma\in(0,1]$, (25) leads to the desired result. $\square$

*Remark* A.9. From (25), a sufficient condition for constants $C_4'$ and $C_5'$ in Lemma A.6 is

$$\sqrt{C_5'}\frac{C_4'}{C_4'\|\boldsymbol{v}\|_\infty + 1} \ge \frac{4\sqrt{2}}{\sqrt{k}\|\boldsymbol{v}\|_\infty},$$

$$\sqrt{C_5'}\frac{C_4'}{C_4'\|\boldsymbol{v}\|_\infty + 1} \ge 8\sqrt{\frac{3}{c_7}},$$

$$C_5'\frac{C_4'}{C_4'\|\boldsymbol{v}\|_\infty + 1} \ge \frac{24}{c_7\sqrt{k}}.$$

It can be simplified as

$$\sqrt{C_5'}\frac{C_4'}{C_4'\|\boldsymbol{v}\|_\infty + 1} \ge 4\sqrt{2}\max\left\{\frac{\sqrt{6}}{\sqrt{c_7}}, \frac{1}{\sqrt{k}\|\boldsymbol{v}\|_\infty}\right\},$$

or

$$\sqrt{C_5'}\frac{C_4'}{C_4' + 1} \ge 4\sqrt{2}\max\left\{\frac{\sqrt{6}}{\sqrt{c_7}}, 1\right\}.$$

*Step 3: Putting everything together.* Now we estimate $\text{dist}(\boldsymbol{v}^0, \boldsymbol{v})$ and prove Theorem 3.2.

*Proof of Theorem 3.2.* For any $\gamma\in(0,1]$, we assume $\lambda \ge C_1\|\boldsymbol{v}\|_\infty^{-1}$ and $n \ge C_2\gamma^{-2}k\log p$ with some absolute constants $C_1, C_2 > 0$. For simplicity, we denote $\rho = \rho(\boldsymbol{E}, k)$. Under the above assumptions, by applying Lemma A.1 with $t = \frac{1}{16}\gamma$, $s = k$ and Lemma A.8, we have

$$\mathbb{P}\left\{\rho \le \frac{3}{16}\gamma\lambda, j_0\in\mathcal{S}_\gamma^+ \subset \hat{\mathcal{S}}\right\} \ge 1 - 2p^{-k} - 3p^{-1} \ge 1 - 5p^{-1}. \tag{26}$$

Under the event in (26), we estimate $\text{dist}(\boldsymbol{v}^0, \boldsymbol{v})$. Since $\text{supp}(\boldsymbol{v}^0) = \hat{\mathcal{S}}$, we have

$$\text{dist}(\boldsymbol{v}^0, \boldsymbol{v})^2 = \text{dist}(\boldsymbol{v}^0, \boldsymbol{v}_{\hat{\mathcal{S}}})^2 + \|\boldsymbol{v}_{\hat{\mathcal{S}}^c}\|_2^2. \tag{27}$$

Firstly, we estimate $\|\boldsymbol{v}_{\hat{\mathcal{S}}^c}\|_2^2$. Since $\hat{\mathcal{S}}^c \subset (S \setminus \mathcal{S}_\gamma^-)^c = \mathcal{S}_\gamma^- \bigcup \mathcal{S}^c$, we have

$$\|\boldsymbol{v}_{\hat{\mathcal{S}}^c}\|_2^2 \leq \|\boldsymbol{v}_{\mathcal{S}_\gamma^-}\|_2^2 + \|\boldsymbol{v}_{\mathcal{S}^c}\|_2^2 < \frac{\gamma^2}{4} < \frac{1}{4}, \ \|\boldsymbol{v}_{\hat{\mathcal{S}}}\|_2^2 > 1 - \frac{\gamma^2}{4} > \frac{3}{4}.$$

Secondly, we estimate $\text{dist}(\boldsymbol{v}^0, \boldsymbol{v}_{\hat{\mathcal{S}}})^2$. Applying Lemma A.3 with $\Lambda = \hat{\mathcal{S}}$ and $s = k$, we obtain

$$\text{dist}(\boldsymbol{v}^0, \boldsymbol{v}_{\hat{\mathcal{S}}})^2 \leq \|\boldsymbol{v}_{\hat{\mathcal{S}}}\|_2 + 1 - 2\frac{\|\boldsymbol{v}_{\hat{\mathcal{S}}}\|_2}{\sqrt{1 + \frac{\rho^2}{[\lambda\|\boldsymbol{v}_{\hat{\mathcal{S}}}\|_2^2 - 2\rho]^2}}} \leq \|\boldsymbol{v}_{\hat{\mathcal{S}}}\|_2 + 1 - 2\frac{\|\boldsymbol{v}_{\hat{\mathcal{S}}}\|_2}{\sqrt{1 + \frac{\rho^2}{(\frac{3}{4}\lambda - 2\rho)^2}}},$$

where the last inequality holds since $\|\boldsymbol{v}_{\hat{\mathcal{S}}}\|_2^2 > \frac{3}{4}$. Therefore, by using Lemma A.3 and $\|\boldsymbol{v}_{\hat{\mathcal{S}}^c}\|_2^2 \leq \frac{\gamma^2}{4}$, we have

$$\text{dist}(\boldsymbol{v}^0, \boldsymbol{v}_{\hat{\mathcal{S}}})^2 \leq \max\left\{ 2 - \frac{\gamma^2}{4} - \frac{2\sqrt{1 - \frac{\gamma^2}{4}}}{\sqrt{1 + \frac{\rho^2}{(\frac{3}{4}\lambda - 2\rho)^2}}}, 2 - \frac{2}{\sqrt{1 + \frac{\rho^2}{(\frac{3}{4}\lambda - 2\rho)^2}}} \right\}$$

$$\leq \max\left\{ 2 - \frac{\gamma^2}{4} - 2\frac{1 - \frac{\gamma^2}{4}}{1 + \frac{\rho^2}{(\frac{3}{4}\lambda - 2\rho)^2}}, 2 - 2\frac{1}{1 + \frac{\rho^2}{(\frac{3}{4}\lambda - 2\rho)^2}} \right\}$$

$$= \max\left\{ \frac{\frac{\gamma^2}{4}(\frac{3}{4}\lambda - 2\rho)^2 + (2 - \frac{\gamma^2}{4})\rho^2}{(\frac{3}{4}\lambda - 2\rho)^2 + \rho^2}, \frac{2\rho^2}{(\frac{3}{4}\lambda - 2\rho)^2 + \rho^2} \right\}$$

$$\leq \frac{\gamma^2}{4} + \frac{2\rho^2}{(\frac{3}{4}\lambda - 2\rho)^2 + \rho^2}.$$

It follows from (27) and $\rho \leq \frac{3}{16}\gamma\lambda$ that

$$\text{dist}(\boldsymbol{v}^0, \boldsymbol{v})^2 \leq \frac{\gamma^2}{2} + \frac{\gamma^2}{2} = \gamma^2,$$

completing the proof. $\qquad\square$

*Remark* A.10. From (13)(25), similar to Remark A.9, a sufficient condition for constants $C_1$ and $C_2$ in Theorem 3.2 is

$$\sqrt{C_2}\frac{C_1}{C_1 + \|\boldsymbol{v}\|_\infty} \geq 16\sqrt{\frac{2 + \log_2(9e)}{C_1'}},$$

$$\sqrt{C_2}\frac{C_1}{C_1\|\boldsymbol{v}\|_\infty + 1} \geq 4\sqrt{2}\max\left\{ \frac{\sqrt{6}}{\sqrt{c_7}}, \frac{1}{\sqrt{k}\|\boldsymbol{v}\|_\infty} \right\},$$

which can be simplified as

$$\sqrt{C_2}\frac{C_1}{C_1 + 1} \geq \max\left\{ 16\sqrt{\frac{2 + \log_2(9e)}{C_1'}}, 4\sqrt{2}\max\left\{ \frac{\sqrt{6}}{\sqrt{c_7}}, 1 \right\} \right\}.$$

## A.4. Proof of Theorem 3.3

We organize the proof of Theorem 3.3 into two parts. First, we show that $v^0$ falls into a small constant neighborhood of $v$. Subsequently, we prove the convergence of the truncated power method.

*Proof of Theorem 3.3.* Throughout the proof, we assume $\lambda \geq C_3 \|v\|_\infty^{-1}$ and $n \geq C_4 k \log p$ with some absolute constants $C_3, C_4 > 0$. We denote $\widetilde{k} = k + 2k'$, $\rho = \rho(E, \widetilde{k})$ and $\mathcal{F}_t = \text{supp}(v^t)$, where $k' = C_5 k$ for some absolute constant $C_5 \geq 1$. Setting $t = \frac{1}{3}\zeta\sqrt{\frac{k \log p}{n}}$ and $s = \widetilde{k}$ in Lemma A.1 and $\gamma = 1$ in Lemma A.8 for some constant $0 < \zeta < 0.001\sqrt{C_4}$, together with the above assumptions, similar to Theorem 3.2, with the probability exceeding $1 - 2p^{-k} - 3p^{-1} \geq 1 - 5p^{-1}$, the following event holds:

$$\mathcal{E} = \left\{ \rho \leq \zeta\sqrt{\frac{k \log p}{n}}\lambda, \ \text{dist}(v^0, v) \leq 1 \right\}.$$

We will continue the proof under this event.

*Step 1: Estimating $|v^T v^0|$.* Since $1 \geq \text{dist}(v, v^0)^2 = 2 - 2|v^T v^0|$, we have $|v^T v^0| \geq 0.5$.

*Step 2: Convergence of truncated power method.* To prove (11), we will first show that $\text{dist}(v, v^t) \leq 1$ by induction.

We denote $\Lambda_t = \mathcal{F}_{t-1} \cup \mathcal{F}_t \cup \mathcal{S}$, then $|\Lambda_t| \leq k + 2k' = \widetilde{k}$. Also, we define

$$u^t = \hat{\Sigma}_{\Lambda_t} v^{t-1} / \|\hat{\Sigma}_{\Lambda_t} v^{t-1}\|_2, \tag{28}$$

hence we have $u^t = u^t_{\mathcal{F}_t} / \|u^t_{\mathcal{F}_t}\|_2$. Let $\kappa$ be the ratio of the second largest (in absolute value) to the largest eigenvalue of $\hat{\Sigma}_{\Lambda_t}$. Then, since $\mathcal{S} \subset \Lambda_t$, similar to (18)(19), we obtain

$$\kappa = \frac{\max_{j \neq 1} \left| \lambda_j(\hat{\Sigma}_{\Lambda_t}) \right|}{\left| \lambda_1(\hat{\Sigma}_{\Lambda_t}) \right|} \leq \frac{1 + \rho}{\lambda\|v_{\Lambda_t}\|_2^2 + 1 - \rho} \leq \frac{1 + 0.001\lambda}{1 + 0.999\lambda} \leq \frac{\|v\|_\infty + 0.001C_3}{\|v\|_\infty + 0.999C_3} =: B_1,$$

where in the second inequality we use $\rho \leq \zeta\sqrt{\frac{k \log p}{n}}\lambda \leq 0.001\lambda$, and in the last inequality we use $\lambda \geq C_3\|v\|_\infty^{-1}$. Obviously we have $\kappa \leq B_1 < 1$.

Let $\bar{v}$ be a unit eigenvector corresponding to the largest eigenvalue of $\hat{\Sigma}_{\Lambda_t}$ and satisfying $v^T \bar{v} \geq 0$. hence we have $\text{dist}(v, \bar{v}) = \|v - \bar{v}\|_2$. Then, using (28) and Lemma A.4, we have

$$\left| \bar{v}^T u^t \right| \geq \left| \bar{v}^T v^{t-1} \right| \left( 1 + \frac{1}{2}(1 - \kappa^2)\left( 1 - \left| \bar{v}^T v^{t-1} \right|^2 \right) \right),$$

which implies that

$$1 - \left| \bar{v}^T u^t \right| \leq \left( 1 - \left| \bar{v}^T v^{t-1} \right| \right) \left( 1 - \frac{1 - B_1^2}{2} \left( \left| \bar{v}^T v^{t-1} \right| + \left| \bar{v}^T v^{t-1} \right|^2 \right) \right). \tag{29}$$

Since $\mathcal{S} \subset \Lambda_t$, Lemma A.3 gives

$$\|v - \bar{v}\|_2^2 = \text{dist}(v, \bar{v})^2 \leq 2 - 2\frac{1}{\sqrt{1 + \frac{\rho^2}{(\lambda - 2\rho)^2}}} \leq \frac{\rho^2}{(\lambda - 2\rho)^2} \leq \frac{\zeta^2}{0.998^2}\frac{k \log p}{n} \leq \frac{1}{998^2}, \tag{30}$$

where in the second inequality we use $1 - \frac{1}{\sqrt{1+a}} \leq \frac{a}{2}$ for $a \geq 0$, and in the last two inequalities we use $\rho \leq \zeta\sqrt{\frac{k \log p}{n}}\lambda \leq 0.001\lambda$ and the conditions of $n$ and $\zeta$. Note that the induction assumption $\text{dist}(v, v^{t-1}) \leq 1$ implies that $|v^T v^{t-1}| \geq 0.5$, which with (30) further leads to

$$\left| \bar{v}^T v^{t-1} \right| \geq \left| v^T v^{t-1} \right| - \left| (v - \bar{v})^T v^{t-1} \right| \geq \left| v^T v^{t-1} \right| - \|v - \bar{v}\|_2 \|v^{t-1}\|_2 \geq 0.4989. \tag{31}$$

Plugging (31) into (29), we have

$$1 - \left| \bar{v}^T u^t \right| \leq \left( 0.6261 + 0.374B_1^2 \right)\left( 1 - \left| \bar{v}^T v^{t-1} \right| \right),$$

which is equivalent to

$$\text{dist}(\bar{\boldsymbol{v}}, \boldsymbol{u}^t) \le \sqrt{0.6261 + 0.374 B_1^2} \cdot \text{dist}(\bar{\boldsymbol{v}}, \boldsymbol{v}^{t-1}), \tag{32}$$

where we use $\|\bar{\boldsymbol{v}}\|_2 = \|\boldsymbol{u}^t\|_2 = \|\boldsymbol{v}^{t-1}\|_2 = 1$. For unit vectors $\bar{\boldsymbol{v}}, \boldsymbol{v}^{t-1}, \boldsymbol{v}$, we obtain

$$\text{dist}(\bar{\boldsymbol{v}}, \boldsymbol{v}^{t-1}) \le \text{dist}(\bar{\boldsymbol{v}}, \boldsymbol{v}) + \text{dist}(\boldsymbol{v}^{t-1}, \boldsymbol{v}). \tag{33}$$

This is because

$$
\begin{aligned}
\text{dist}(\bar{\boldsymbol{v}}, \boldsymbol{v}) + \text{dist}(\boldsymbol{v}^{t-1}, \boldsymbol{v}) &= \|\tau_1 \bar{\boldsymbol{v}} - \boldsymbol{v}\|_2 + \|\boldsymbol{v} + \tau_2 \boldsymbol{v}^{t-1}\|_2 \\
&\ge \|\tau_1 \bar{\boldsymbol{v}} + \tau_2 \boldsymbol{v}^{t-1}\|_2 \\
&\ge \text{dist}(\bar{\boldsymbol{v}}, \boldsymbol{v}^{t-1}),
\end{aligned}
$$

where $\tau_1, \tau_2 \in \{\pm 1\}$ and we use (10). Similarly, for unit vectors $\boldsymbol{v}, \boldsymbol{u}^t, \bar{\boldsymbol{v}}$, it holds that

$$\text{dist}(\boldsymbol{v}, \boldsymbol{u}^t) \le \text{dist}(\boldsymbol{v}, \bar{\boldsymbol{v}}) + \text{dist}(\boldsymbol{u}^t, \bar{\boldsymbol{v}}). \tag{34}$$

Using (30), (32), (33), and (34), we have

$$
\begin{aligned}
&\text{dist}(\boldsymbol{v}, \boldsymbol{u}^t) \\
&\le \sqrt{0.6261 + 0.374 B_1^2} \cdot \text{dist}(\boldsymbol{v}, \boldsymbol{v}^{t-1}) + \frac{\zeta}{0.998}(1 + \sqrt{0.6261 + 0.374 B_1^2}) \sqrt{\frac{k \log p}{n}}.
\end{aligned} \tag{35}
$$

Since $k' = C_5 k$, Lemma A.5 generates

$$
\begin{aligned}
\left| \boldsymbol{v}^T [\boldsymbol{u}^t]_{\mathcal{F}_t} \right| &\ge \left| \boldsymbol{v}^T \boldsymbol{u}^t \right| - C_5^{-1/2} \min \left\{ \sqrt{1 - |\boldsymbol{v}^T \boldsymbol{u}^t|^2}, (1 + C_5^{-1/2}) \left( 1 - |\boldsymbol{v}^T \boldsymbol{u}^t|^2 \right) \right\} \\
&\ge \left| \boldsymbol{v}^T \boldsymbol{u}^t \right| - C_5^{-1/2}(1 + C_5^{-1/2}) \left( 1 - |\boldsymbol{v}^T \boldsymbol{u}^t|^2 \right),
\end{aligned}
$$

which implies that

$$
1 - \left| \boldsymbol{v}^T [\boldsymbol{u}^t]_{\mathcal{F}_t} \right| \le 1 - \left| \boldsymbol{v}^T \boldsymbol{u}^t \right| + C_5^{-1/2}(1 + C_5^{-1/2}) \left( 1 - |\boldsymbol{v}^T \boldsymbol{u}^t|^2 \right) \le B_2^2 (1 - |\boldsymbol{v}^T \boldsymbol{u}^t|),
$$

where $B_2 := \sqrt{1 + 2C_5^{-1/2}(1 + C_5^{-1/2})}$. Recall that $\boldsymbol{v}^t = \boldsymbol{u}^t_{\mathcal{F}_t} / \|\boldsymbol{u}^t_{\mathcal{F}_t}\|_2$. Then we have

$$
\begin{aligned}
&\text{dist}(\boldsymbol{v}, \boldsymbol{v}^t) \\
&= \sqrt{2 - 2 |\boldsymbol{v}^T \boldsymbol{v}^t|} = \sqrt{2 - 2 |\boldsymbol{v}^T [\boldsymbol{u}^t]_{\mathcal{F}_t}| / \|[\boldsymbol{u}^t]_{\mathcal{F}_t}\|_2} \\
&\le \sqrt{2 - 2 |\boldsymbol{v}^T [\boldsymbol{u}^t]_{\mathcal{F}_t}|} \le B_2 \cdot \sqrt{2(1 - |\boldsymbol{v}^T \boldsymbol{u}^t|)} \\
&= B_2 \cdot \text{dist}(\boldsymbol{v}, \boldsymbol{u}^t) \\
&\le B_2 \sqrt{0.6261 + 0.374 B_1^2} \cdot \text{dist}(\boldsymbol{v}, \boldsymbol{v}^{t-1}) + \frac{\zeta B_2}{0.998} \left( 1 + \sqrt{0.6261 + 0.374 B_1^2} \right) \sqrt{\frac{k \log p}{n}} \\
&\le B_2 \sqrt{0.6261 + 0.374 B_1^2} \cdot \text{dist}(\boldsymbol{v}, \boldsymbol{v}^{t-1}) + \frac{B_2}{998} \left( 1 + \sqrt{0.6261 + 0.374 B_1^2} \right),
\end{aligned} \tag{36}
$$

where in the last second inequality we use (35), and the last inequality holds from the conditions of $n$ and $\zeta$. Since $\text{dist}(\boldsymbol{v}, \boldsymbol{v}^{t-1}) \le 1$, the above inequality also implies that $\text{dist}(\boldsymbol{v}, \boldsymbol{v}^t) \le 1$ with suitable constants $C_3, C_5$ (constants $B_1, B_2$). Therefore, we complete the induction, which proves that $\text{dist}(\boldsymbol{v}, \boldsymbol{v}^t) \le 1$ for all $t$. As a result, the above inequality holds for all $t$, which leads to

$$
\begin{aligned}
\text{dist}(\boldsymbol{v}, \boldsymbol{v}^t) &= d \cdot \text{dist}(\boldsymbol{v}, \boldsymbol{v}^{t-1}) + B_3 \sqrt{\frac{k \log p}{n}} \\
&\le d^2 \cdot \text{dist}(\boldsymbol{v}, \boldsymbol{v}^{t-2}) + d \cdot B_3 \sqrt{\frac{k \log p}{n}} + B_3 \sqrt{\frac{k \log p}{n}} \\
&\le \cdots \\
&\le d^t \cdot \text{dist}(\boldsymbol{v}, \boldsymbol{v}^0) + \frac{B_3}{1 - d} \sqrt{\frac{k \log p}{n}},
\end{aligned}
$$

where $d := B_2\sqrt{0.6261 + 0.374B_1^2}$ and $B_3 := \frac{\zeta B_2}{0.998}\left(1 + \sqrt{0.6261 + 0.374B_1^2}\right)$. Denoting $d' := \frac{B_3}{1-d}$, the above inequality is the desired result. $\qquad\square$

*Remark* A.11. From (13)(25)(36), similar to Remark A.10, a sufficient condition for constants $C_3, C_4, C_5, \zeta$ in Theorem 3.3 is

$$\zeta \le 0.001\sqrt{C_4},$$

$$\frac{\zeta}{\sqrt{C_5}}\frac{C_3}{C_3 + \|v\|_\infty} \ge \frac{1}{3}\sqrt{\frac{4 + 3\log_2(3e)}{3C_1'}},$$

$$\sqrt{C_4}\frac{C_3}{C_3\|v\|_\infty + 1} \ge 4\sqrt{2}\max\left\{\frac{\sqrt{6}}{\sqrt{c_7}}, \frac{1}{\sqrt{k}\|v\|_\infty}\right\},$$

$$999B_2\sqrt{0.6261 + 0.374B_1^2} + B_2 \le 998,$$

where $B_1 = \frac{\|v\|_\infty + 0.001C_3}{\|v\|_\infty + 0.999C_3}$ and $B_2 = \sqrt{1 + 2C_5^{-1/2}(1 + C_5^{-1/2})}$. It can be simplified as

$$\zeta \le 0.001\sqrt{C_4},$$

$$\frac{\zeta}{\sqrt{C_5}}\frac{C_3}{C_3 + 1} \ge \frac{1}{3}\sqrt{\frac{4 + 3\log_2(3e)}{3C_1'}},$$

$$\sqrt{C_4}\frac{C_3}{C_3 + 1} \ge 4\sqrt{2}\max\left\{\frac{\sqrt{6}}{\sqrt{c_7}}, 1\right\},$$

$$999B_2\sqrt{0.6261 + 0.374\widetilde{B_1}^2} + B_2 \le 998,$$

where $\widetilde{B_1} = \frac{1 + 0.001C_3}{1 + 0.999C_3}$.

# B. Additional Experiments

To further validate and analyze the behavior of our proposed Algorithm 2, we conduct two complementary studies. First, we assess the robustness of the algorithm under suboptimal signal-strength conditions by varying the parameter $\lambda$, using the same data-generation procedure as in Section 4.2. Second, we isolate the contribution of the refinement stage by comparing the estimation accuracy after the initialization step alone with that of the full two-stage pipeline.

## B.1. Robustness of Algorithm 2

To assess the robustness of Algorithm 2 when theoretical assumptions are relaxed, we conduct experiments across a range of signal strength values $\lambda$. Following the experimental protocol from Section 4.2, we generate sparse eigenvectors $\boldsymbol{v}$ with dimension $p = 1000$ and sparsity level $k = 20$. All reported results represent averages over 200 independent trials.

*Table 4.* Estimation error verus sample size $n$ under different $\lambda$

| Sample size $n$ | 100 | 200 | 300 | 400 | 500 | 600 | 700 | 800 | 900 | 1000 |
|---|---|---|---|---|---|---|---|---|---|---|
| $\lambda = 0.5$ | 1.4003 | 1.3971 | 1.3973 | 1.3996 | 1.3976 | 1.3903 | 1.3939 | 1.3943 | 1.3927 | 1.3897 |
| $\lambda = 2.5$ | 1.3174 | 1.1650 | 0.7896 | 0.5427 | 0.3980 | 0.2459 | 0.1736 | 0.1535 | 0.1331 | 0.1280 |
| $\lambda = 5$ | 0.7188 | 0.2635 | 0.1347 | 0.1062 | 0.0943 | 0.0863 | 0.0789 | 0.0751 | 0.0701 | 0.0664 |
| $\lambda = 7.5$ | 0.2762 | 0.1246 | 0.0961 | 0.0843 | 0.0749 | 0.0683 | 0.0626 | 0.0596 | 0.0556 | 0.0526 |

Table 4 presents the estimation error as a function of sample size for varying signal strengths. The results show that larger $\lambda$ values yield substantially lower estimation errors, and that the advantage of strong-signal settings over weak-signal settings becomes increasingly pronounced as $n$ grows. Crucially, even under weak signals (small $\lambda$), the algorithm degrades gracefully—maintaining robust accuracy rather than failing catastrophically. These findings underscore the practical resilience of our method across a broader range of scenarios than our theoretical guarantees alone would cover.

## B.2. Improvement of Refinement Stage in Algorithm 2

To quantify the contribution of the refinement stage in Algorithm 2, we compare the estimation accuracy after the initialization stage alone versus the complete two-stage procedure. We maintain the experimental protocol from Section 4.2 for generating $\boldsymbol{v}$, setting dimension $p = 1000$ and sparsity level $k = 20$. Results are averaged over 200 independent trials.

*Table 5.* Estimation error verus sample size $n$ after different stages in Algorithm 2

| Sample size $n$ | 100 | 200 | 300 | 400 | 500 | 600 | 700 | 800 | 900 | 1000 |
|---|---|---|---|---|---|---|---|---|---|---|
| Initialization | 1.0553 | 0.6733 | 0.4183 | 0.2880 | 0.1981 | 0.1401 | 0.1028 | 0.0848 | 0.0750 | 0.0697 |
| Refinement | 0.7188 | 0.2635 | 0.1347 | 0.1062 | 0.0943 | 0.0863 | 0.0789 | 0.0751 | 0.0701 | 0.0664 |

Table 5 reveals the substantial improvement achieved by the refinement stage. The truncated power iteration consistently reduces estimation error across all sample sizes, with the most dramatic gains occurring at smaller sample sizes—for instance, reducing error by over 60% when $n = 200$. As the sample size increases, both stages converge toward similar performance, yet the refinement stage maintains a consistent advantage. These empirical findings validate our two-stage design and demonstrate that combining careful initialization with iterative refinement is essential for achieving optimal statistical performance.

