# OpenReview forum: "Fast and Provable Algorithms for Sparse PCA with Improved Sample Complexity"
_ICML.cc/2025/Conference — ICML 2025 poster_

### Official Review · Reviewer_z5wB · 2025-03-08

**Overall Recommendation:** 3

**Summary:**

The paper proposes a two-stage algorithm to obtain the principal component of the single-spiked covariance model (Sparse PCA). The first stage, called the thresholding-based algorithm, obtains a first estimation of the principal component and, most importantly, accomplishes reduced computational cost compared to competing algorithms such as diagonal thresholding. In particular, it reduces the number of data samples required for the computation of the principal component – assuming a restricted strength of the signal of the single-spiked covariance model. To further enhance this estimation, the second stage utilizes the truncated power iteration to refine the solution, accomplishing minmax optimal rate.

UPDATE AFTER REBUTTAL
Thanks for replying to all the questions. Most of them were properly addressed, and I appreciate the effort to clarify both theoretical and empirical aspects. However, I would prefer that the answers which mention future inclusion of additional experiments or material in the final version already present that content within the rebuttal. Promising to include something later is not entirely sufficient, as it leaves the possibility that the addition may not be implemented after acceptance. Including this material now would allow reviewers to properly assess its quality and relevance during the review process. In any case, after reading the rest of the comments and the corresponding response, I would like to update my final score.

**Claims And Evidence:**

Overall, the paper mainly provides proper theoretical proves and derivations to theoretically demonstrate the computational complexity of the algorithm and its theoretical error. Additionally, it evaluates with synthetic data the main theoretical claims; one is missing regarding diagonal thresholding and I also miss experiments regarding the two stage refinement (see questions for authors).

**Essential References Not Discussed:**

Main references are included.

**Experimental Designs Or Analyses:**

I have checked the experimental analysis. From my perspective, experiments are properly aligned with the paper, but two main questions arises from two experiments that lack: one is missing regarding diagonal thresholding and I also miss experiments regarding the two stage refinement (see questions for authors).

**Methods And Evaluation Criteria:**

Overall, the paper mainly provides proper theoretical proves and derivations to theoretically demonstrate the computational complexity of the algorithm and its theoretical error. Additionally, it evaluates with synthetic data the main theoretical claims; one is missing regarding diagonal thresholding and I also miss experiments regarding the two stage refinement (see questions for authors).

**Other Comments Or Suggestions:**

3) "reduce the parameter space of the model" --> include here the notation that you use
4) second stage incurs O(np) costs.. --> typo, two dots
5) 2.1. Preliminaries --> explain you work with zero-mean observations
6) to estimate a single sparse vector v, to identify a sparse unit vector w --> explain better the differences between v and w
7) This approach is based on the observation that the expectation values of the diagonal entries of --> missing a reference to Eq. 2
8) Is v^~ notation introduced in Equation (8)?
9) When you define the Error in Section 4, you are repeating the same equation than Eq. 10, just reference it. Moreveor, this may be just a styling thing, but Error and F-score appear in another font and font size, which does not look really formal for a paper.

**Other Strengths And Weaknesses:**

Overall, I think that the paper is good, but it requires addressing a few things to be a clearly ICML publcication. Please address the two comments in “Other Strengths And Weaknesses” and the three questions in “Questions For Authors”. If they are addressed, I would modify my recommendation to accept:
1) I would rather start with an introductory text explaining the single-spiked covariance model, instead of just starting with the mathematical formulation; also missing including the citation (Johnstone, 2001). Moreover, for me, it is missing in this introductory text part the basic differences between PCA and sparse PCA/single-spiked covariance model: we assume that the data can be explained with a unique principal component, coefficients follow a standard normal (instead of are "sampled"), epsilon_i gathers the inherent noise or the information not explained by the single component, an example or two of applications/problems where this model is useful, etc.
2) Sorry if I miss it in the paper, but I think it would be interesting to either include the prove that diagonal thresholding (Johnstone & Lu, 2009) requires n=Ω (k 2 logp) or either to include a discussion/counterexample illustrating that diagonal thresholding does not fulfill Theorem 3.2 (maybe this will be ever more interesting, as the other one is included in the original paper). This will showcase that your algorithm is effectively better than diagonal thresholding. It would be ok to prove it in the appendix and just reference it with one sentence.

**Questions For Authors:**

Overall, I think that the paper is good, but it requires addressing a few things to be a clearly ICML publcication. Please address the two comments in “Other Strengths And Weaknesses” and the three questions in “Questions For Authors”. If they are addressed, I would modify my recommendation to accept:
10) You mention that "operating under a specific condition on λ", which is the implication of this restriction? It would be interesting to include it this discussion in the paper, either theoretically or experimentally.
11) I understand that the two-stage algorithms theoretically guarantees the minimax optimal rate. However, until which level is this refinement necessary? In other words, until which level refines the your proposed thresholding algorithm? This is missing. For instance, include an experiment showing before and after the refinement using your Error metric.
12) I understood that Diagonal Thresholding (DT) is computationally slower than your proposed algorithm. However, in Figure 2 b, we see that it is as fast as yours. Then, if we include the refinement using the Truncated power method (3.1.2), then we could accomplish the same performance with the same computationally efficiency?

**Relation To Broader Scientific Literature:**

The key contributions of this paper build upon previous work in sparse PCA, particularly in improving sample complexity and computational efficiency. By introducing a novel thresholding algorithm for principal component initialization and truncated power iteration, the paper provides an efficient solution that achieves optimal statistical guarantees with significantly reduced sample complexity.

**Theoretical Claims:**

Theoretical proves are included in the supplementary, which I have not reviewed.

---

> ### Author Rebuttal · Authors · 2025-04-01
>
> > 1. You mention that "operating under a specific condition on $\lambda$", which is the implication of this restriction? It would be interesting to include it this discussion in the paper, either theoretically or experimentally.
>
> **Reply:** **Please refer to the response to the first question for Reviewer gGz9 for a detailed discussion on the strength and necessity of this condition.**
>
>
> > 2. Include an experiment showing before and after the refinement using your error metric.
>
> **Reply:** Thank you for your question. Our experimental results clearly demonstrate that the refinement stage significantly enhances the performance of the initial estimator. In the table below, you can see that the estimation error is consistently reduced after applying the refinement across various sample sizes:
>
> | Sample Size **n** | 100    | 200    | 300    | 400    | 500    | 600    | 700    | 800    | 900    | 1000   |
> |-------------------|--------|--------|--------|--------|--------|--------|--------|--------|--------|--------|
> | **Initialization**   | 1.0553 | 0.6733 | 0.4183 | 0.2880 | 0.1981 | 0.1401 | 0.1028 | 0.0848 | 0.0750 | 0.0697 |
> | **Refinement**   | 0.7188 | 0.2635 | 0.1347 | 0.1062 | 0.0943 | 0.0863 | 0.0789 | 0.0751 | 0.0701 | 0.0664 |
>
> These results indicate that while the initial (thresholding) algorithm provides a solid baseline, the refinement stage significantly reduces the estimation error. In the final version, we will include additional experiments to compare the performance before and after refinement.
>
> > 3. I understood that Diagonal Thresholding (DT) is computationally slower than your proposed algorithm. However, in Figure 2(b), we see that it is as fast as yours. Then, if we include the refinement using the Truncated power method (3.1.2), then we could accomplish the same performance with the same computationally efficiency?
>
> **Reply:** Thank you for your question. As explained in Section 3.2, both Diagonal Thresholding (DT) and our initialization algorithm incur the same order of computational cost of $O(np + nk^2)$, since they involve computing the entries of the sample covariance matrix and performing an eigenvalue decomposition.
>
> Our refinement stage utilizes truncated power iterations, each costing $O(np)$. In our experiments (see Tables 1-3), the number of iterations required for convergence is very small---typically no more than 10 iterations. Consequently, because the initialization stage dominates the overall cost, the additional refinement improves accuracy without compromising computational efficiency. This is further supported by Figure 2(b), where our two-stage method (red line) runs as fast as DT (green line).
>
> > 4.  Start with an introductory text explaining the single-spiked covariance model. It is missing the basic differences between PCA and sparse PCA.
>
> **Reply:** Thank you for your valuable suggestion. In the final version of our manuscript, we plan to revise the introductory text in Section 1 to improve clarity and provide a more comprehensive overview of the single-spiked covariance model. Specifically, we will:
>
> - Introduce principal component analysis (PCA) by emphasizing its importance, typical applications, and inherent limitations (e.g., the production of dense principal components).
>
> - Present sparse PCA as a remedy to these limitations by highlighting its role in generating more interpretable components.
>
> - Provide a detailed explanation of the single-spiked covariance model and add the citation (Johnstone, 2001) when presenting the mathematical formulation.
>
> - Include one or two examples of applications or problems where this model proves particularly useful.
>
> We believe that these revisions will enhance the clarity and accessibility of the introductory section.
>
> >5.  Why diagonal thresholding does not fullfill Theorem 3.2.
>
> **Reply:** Thank you for your comment. With assumption $\lambda = \Omega({||v||} _{\infty}^{-1})$, diagonal thresholding (DT) cannot achieve sample complexity $\Omega(k \log p)$. This is based on the following observations and analysis.
>
> The sample complexity of DT is governed by the statistical gap defined in Eq.(5) of our paper. In contrast, the sample complexity of Algorithm 1 is governed by the statistical gap as Eq.(7) in our paper. A larger gap leads to a smaller sample complexity. If the infinity norm of $v$ is of constant order, then our gap depends only linearly on $\min_{j \in S} |v_j|$, improving order-wisely the gap of DT and consequently reducing the sample complexity $\Omega(k^2 \log p)$ to $\Omega(k \log p)$. However, for DT, the constant order of ${||v||} _{\infty}$ does not improve its gap, suggesting that its sample complexity cannot be improved.
>
> In conclusion, Algorithm 1 provides a way to incorporate the property of $v$ into estimation, while DT cannot do it according to our analysis. We will add this discussion in the final version.
>
> > 6. Other Comments:
>
> **Reply:** Thank you—we'll fix the typos.

---

### Official Review · Reviewer_gGz9 · 2025-03-14

**Overall Recommendation:** 3

**Summary:**

This paper presents their algorithms for Sparse Principal Component Analysis (Sparse PCA) under the setting of the Single-Spiked Covariance Model. With the assumption of signal strength, the authors introduce a thresholding-based algorithm with better (big-Omega) sample complexity and show it merges the gap of existing polynomial-time algorithms and information-theoretic lower bounds. Additionally, they propose a two-stage nonconvex optimization algorithm that refines the estimation using truncated power iteration, achieving minimax-optimal statistical error rates. The author provided theoretical analysis for the proposed algorithms and showed the numerical experiments that verified their performance claim in terms of estimation accuracy, sample complexity, and computational cost.

## update after rebuttal

I have read the author's response and would like to thank them for their detailed rebuttal. After reading other reviewers' opinions and the rebuttal, I acknowledge that the proposed method demonstrates a reasonable degree of practical robustness, even though the underlying assumptions may be somewhat idealized. Hence, I have decided to maintain my original score of Weak Accept.

**Claims And Evidence:**

The reviewer is not from the community of sparse PCA. The reviewer is convinced by the claims in general but some concerns need to be justified.

From intuition, the ‘Single-Spiked Covariance’ may be not suitable for some cases with multiple principle components, which the analysis will not work for sure. It seems quite strong combining the signal strength assumption. Could the author have explained and verified this is a common setting in sparse PCA?

Besides, it assumes i.i.d. Gaussian Noise, is there other cases that this is not satisfied and then your analysis fails? How does it work in more real applications?

What is the robustness of different cases of SNR?

**Essential References Not Discussed:**

N/A

**Experimental Designs Or Analyses:**

The synthetic experiments are based on their tailored designed setting, this is fine since it's hard to find ground truth for sparse PCA. However, it may be not practical since the assumption is quite strong.

Also, the reviewer would suggest verifying the proposed method for some real applications that heavily rely on the sample complexity, computational cost, and accuracy.

**Methods And Evaluation Criteria:**

The synthetic experiments are based on their tailored designed setting, this is fine since it's hard to find ground truth for sparse PCA. However, it may be not practical since the assumption is quite strong.

Also, the reviewer would suggest verifying the proposed method for some real applications that heavily rely on the sample complexity, computational cost, and accuracy.

**Other Comments Or Suggestions:**

N/A

**Other Strengths And Weaknesses:**

The truncated power method for sparse eigenvalue problems has already been published. Could the author clarify what is their unique contribution in this part?

**Questions For Authors:**

See my previous discussion.

**Relation To Broader Scientific Literature:**

The reviewer believes this is an interesting work with significant improvement. However, the author needs to justify how general is their algorithm in more practice scenarios in the broader areas.

**Theoretical Claims:**

Under their assumption, the result looks correct and makes sense to the reviewer.

---

> ### Author Rebuttal · Authors · 2025-03-31
>
> > **1. The strength and necessity of the additional assumption on $\lambda$**
>
> **Reply:** The single-spiked covariance model is a well-studied framework in high-dimensional statistics. Nevertheless, there remains a substantial gap between the information-theoretic sample complexity, $\Omega(k \log p)$, and the best known sample complexity of existing polynomial-time algorithms, $\Omega(k^2)$.
>
> **To bridge this gap under the planted clique conjecture, it is necessary to introduce additional assumptions**. In fact, prior work—based on reductions from the planted clique conjecture—provides strong evidence that without extra conditions no polynomial-time algorithm can recover the spike with sample complexity $\Omega(k \log p)$ [R1-R5].
>
> Our assumption $\lambda = \Omega(||v|| _\infty^{-1})$ ensures that the nonzero entries of the spike $v$ decay sufficiently fast. In many theoretical analyses, $\lambda$ is taken to be a constant when deriving both the information-theoretic lower bounds and the sample complexities of various algorithms. Moreover, **in models where the nonzero entries of $v$ follow a power-law decay—a scenario well-known in compressed sensing—$\lambda$ naturally remains of constant order**.
>
> A similar phenomenon was observed in the sparse phase retrieval problem [R6], where signals with power-law decay could be recovered with optimal sample complexity.
>
> We will expand upon this discussion in the revised manuscript.
>
> Reference:
>
> [R1] Statistical and computational trade-offs in estimation of sparse principal components, Annals of Statistics, 2016.
>
> [R2] Do semidefinite relaxations solve sparse PCA up to the information limit, Annals of Statistics, 2015.
>
> [R3] Optimal detection of sparse principal components in high dimension, Annals of Statistics, 2013.
>
> [R4] Sparse CCA: Adaptive estimation and computational barriers, Annals of Statistics, 2017.
>
> [R5] Reducibility and computational lower bounds for problems with planted sparse structure, COLT, 2018.
>
> [R6] Sample-efficient algorithms for recovering structured signals from magnitude-only measurements, IEEE Transactions on Information Theory, 2019.
>
>
> > 2. It assumes i.i.d. Gaussian Noise, is there other cases that this is not satisfied and then your analysis fails? How does it work in more real applications?
>
> **Reply:** Thank you for your question. The single-spiked covariance model is a well-known model in high-dimensional statistic analysis. Its research is mainly from a theoretical perspective. In the single-spiked model, for the sample $x_i = \sqrt{\lambda} g_i v + \xi_i$, the noise $\xi_i$ is generally assumed to follow a standard Gaussian distribution $N(0,I)$, or sometimes it is assumed to follow a sub-Gaussian distribution with variance proxy $O(1)$ [R7]. Therefore, in our paper, we also assume $\xi_i \sim N(0,1)$, which is the same as most of the previous work.
>
> In our theoretical analysis, the concentration inequalities (Line 607-610 in Lemma A.1, Eq.(19) in Lemma A.6, Eq.(22)(23) in Lemma A.8) hold based on this assumption of $\xi_i$. If we assume $\xi_i$ follows a sub-Gaussian distribution with variance proxy $O(1)$ (a generalized Gaussian), we can still derive similar concentration inequalities to complete our analysis. However, if $\xi_i$ is assumed to follow other kinds of distributions, the same or similar concentration inequalities may not hold, and thus the analysis may fail.
>
> We will add this discussion in the revised manuscript.
>
> Reference:
>
> [R7] Sum-of-squares lower bounds for sparse PCA, Advances in Neural Information Processing Systems, 2015.
>
>
> > 3. What is the robustness of different cases of SNR?
>
> **Reply:** **Please refer to the response to the fourth question for Reviewer NHCS for a detailed discussion on the robustness of our method under various SNR regimes.**
>
> > 4. The truncated power method for sparse eigenvalue problems has already been published. Could the author clarify what is their unique contribution in this part?
>
> **Reply:** While the truncated power method was introduced in [R8] and its convergence shown under a good initialization, that work does not provide a practical procedure for obtaining such an initialization nor does it establish the optimal sample complexity. Our unique contribution is the development of a novel thresholding algorithm that produces an initialization satisfying the necessary conditions for the truncated power iterations. This practical initialization enables our two-stage algorithm to achieve the optimal sample complexity of $\Omega(k \log p)$, a significant improvement over the $\Omega(k^2)$ samples typically required by existing polynomial-time methods. In short, our work bridges the gap between theoretical conditions and practical implementation, thereby enhancing both the statistical and computational efficiency of sparse PCA.
>
> Reference:
>
> [R8] Truncated power method for sparse eigenvalue problems, Journal of Machine Learning Research, 2013.

---

### Official Review · Reviewer_NHCS · 2025-03-14

**Overall Recommendation:** 4

**Summary:**

The paper presents efficient algorithms for solving the sparse Principal Component Analysis (PCA) problem, it is one of fundamental problems in machine learning. The proposed algorithm significantly reduces the required sample complexity compared to previous polynomial-time methods. Under typical sparse conditions, prior polynomial-time algorithms required sample complexity on the order of  $O(k^2)$ , whereas this paper achieves near-optimal complexity of $k\log p$, aligning closely with theoretical lower bounds. The authors propose a thresholding-based algorithm along with a two-stage nonconvex approach, combining thresholding initialization with truncated power iteration to ensure both theoretical rigor and computational efficiency. Rigorous proofs demonstrate that the proposed methods significantly improve upon existing polynomial-time methods in terms of both runtime and required sample size. Experiments on synthetic data illustrate practical effectiveness, validating the theoretical claims regarding improved scalability and accuracy.

## update after rebuttal
Thank you for the authors’ responses. I maintain my rating.

**Claims And Evidence:**

The claims presented in this submission are generally supported by the provided evidence.

**Essential References Not Discussed:**

NA

**Experimental Designs Or Analyses:**

The authors utilized synthetic data to examine the trade-offs between solution quality (near-optimality) and computational effort, highlighting the advantages of their proposed algorithms over state-of-the-art methods in achieving a balance between these competing factors.

**Methods And Evaluation Criteria:**

The proposed methods and evaluation criteria are suitable and aligned with the problem addressed in the paper.

**Other Comments Or Suggestions:**

Some numerical experimental results for Algorithm 2 should be moved to the main paper.

**Other Strengths And Weaknesses:**

Strengths:

1) Handling $\ell_0$-constrained sparsity PCA is a challenging task. The paper makes a significant contribution by bridging the gap between the optimal information-theoretic sample complexity of Ω(k log p) and the higher complexities, typically Ω(k²), required by earlier polynomial-time approaches. The authors rigorously support their claims with novel theoretical results and numerical experiments—specifically Theorems 3.2 and 3.3—that establish robust performance guarantees under reasonable assumptions.

2) The innovative integration of thresholding-based initialization with refined truncated power iteration represents a creative approach. This two-stage framework not only enhances statistical performance but also preserves computational efficiency, a crucial factor in high-dimensional settings. A theoretical analysis for the combined algorithm is provided.

Weaknesses:

1) The condition on signal strength that $\lambda = \Omega(\||v\||^{-1}_\infty)$ is crucial for the proposed improvements in sample complexity. Nonetheless, the paper could be enhanced by a more thorough examination of how practical this assumption is. It would be particularly useful to investigate if this condition is frequently met in real-world scenarios or datasets. Furthermore, conducting empirical studies or discussing how robust the method is when this condition is not fully met could significantly enrich the paper.

2) All experiments for the two proposed algorithms are performed using synthetic data. Although these synthetic experiments are crucial for validating theoretical guarantees, demonstrating the algorithms’ performance on real-world, high-dimensional datasets would greatly enhance the practical relevance of the research.

**Questions For Authors:**

1) In Lines 276-279, the authors wrote, “We present a series of numerical experiments designed to verify the theoretical results and validate the efficiency and effectiveness of our proposed two-stage algorithm.” Was Algorithm 1 used in Section 4?

2) In Algorithm 1, when selecting the top $k$ elements, how do you handle cases where some of the last elements are equal? Does this affect the theoretical analysis?

3) What is the stopping criterion for Algorithm 2 in Tables 1-3 in Appendix B? Additionally, how many samples were used for warm-starting (i.e., running the initialization stage)?

**Relation To Broader Scientific Literature:**

The paper presents a significant advancement in the design and analysis of sparse PCA algorithms by reducing sample complexity while maintaining computational efficiency.

**Theoretical Claims:**

I took a glimpse at the proofs; they seem to be correct.

---

> ### Author Rebuttal · Authors · 2025-03-31
>
> > 1. Was Algorithm 1 used in Section 4?
>
> **Reply:** Thank you for your comment. In Section 4, all experiments are conducted using Algorithm 2. This is because Algorithm 2 is our full two-stage procedure, which first employs Algorithm 1 for initialization and then refines the estimate via truncated power iteration. Since the refinement stage in Algorithm 2 consistently improves upon the initialization provided by Algorithm 1, we report only the results from Algorithm 2 in the main experimental section.
>
> > 2. In Algorithm 1, when selecting the top $k$ elements, how do you handle cases where some of the last elements are equal? Does this affect the theoretical analysis?
>
> **Reply:** Thank you for your question. We address the issue in two parts:
>
> **1. Potential Ties Do Not Affect Our Theoretical Analysis:**
>
> In our theoretical analysis (e.g., see Lemma A.8), we establish that with high probability there is a strict separation between the values corresponding to the indices in the true support and those outside of it. Specifically, the minimum value among the selected entries is shown to be strictly greater than the maximum value among the unselected ones. This strict inequality guarantees correct support recovery and renders any ambiguities in the ordering of lower-ranked elements irrelevant. In other words, even if ties were to occur among the lower-ranked entries, they would not compromise the validity of our theoretical guarantees.
>
> **2. The Probability That Any Two Entries Are Exactly Equal Is Zero:**
>
> The values used in our selection process are computed from continuous random variables (i.e., the entries in the empirical covariance matrix derived from i.i.d. samples). In any continuous distribution, the probability that any two independently drawn real numbers are exactly equal is zero, so the occurrence of two identical values is of zero probability.
>
> In summary, our support recovery analysis is robust: it relies on a high-probability strict separation that ensures correct selection, and the continuity of the underlying distributions guarantees that exact ties effectively never occur. We will add this discussion to the revised paper.
>
> > 3. What is the stopping criterion for Algorithm 2 in Tables 1-3 in Appendix B? Additionally, how many samples were used for warm-starting (i.e., running the initialization stage)?
>
> **Reply:** Thank you for your comment. First, the stopping criterion for Algorithm 2 in Tables 1-3 in Appendix B is that the relative error between $v^{t-1}$ and $v^t$ is less than $10^{-8}$. Second, Algorithm 2 employs the same set of samples for both the initialization and refinement stages. Hence, the samples used in the warm-start (initialization stage) are exactly the same as those used in the entire experiment. For example, in the experiments reported in Table 1 of Appendix B, the total sample size is 2500; therefore, 2500 samples are used to warm-start the algorithm.
>
> >  4. Conducting empirical studies or discussing how robust the method is when this condition is not fully met could significantly enrich the paper.
>
> **Reply:** Thank you for your comment. To assess the robustness of our method when the ideal condition is not fully met, we conducted experiments with various values of $ \lambda $. The results below (with $p = 1000$ and $k = 20$) report the estimation error as a function of the sample size:
>
> | Sample size $n$  | 100    | 200    | 300    | 400    | 500    | 600    | 700    | 800    | 900    | 1000   |
> |------------------|--------|--------|--------|--------|--------|--------|--------|--------|--------|--------|
> | $\lambda = 0.5$  | 1.4003 | 1.3971 | 1.3973 | 1.3996 | 1.3976 | 1.3903 | 1.3939 | 1.3943 | 1.3927 | 1.3897 |
> | $\lambda = 2.5$  | 1.3174 | 1.1650 | 0.7896 | 0.5427 | 0.3980 | 0.2459 | 0.1736 | 0.1535 | 0.1331 | 0.1280 |
> | $\lambda = 5$    | 0.7188 | 0.2635 | 0.1347 | 0.1062 | 0.0943 | 0.0863 | 0.0789 | 0.0751 | 0.0701 | 0.0664 |
> | $\lambda = 7.5$  | 0.2762 | 0.1246 | 0.0961 | 0.0843 | 0.0749 | 0.0683 | 0.0626 | 0.0596 | 0.0556 | 0.0526 |
>
> These results show that the estimation error decreases with increasing sample size and that the method remains robust across different $ \lambda $ values, even under less than ideal conditions. We will add this discussion to the revised paper.

---

### Official Review · Reviewer_LyQJ · 2025-03-14

**Overall Recommendation:** 2

**Summary:**

This paper looks at sparse PCA with few samples in the spiked Gaussian
model, under the assumption that the largest single coordinate of the
spike has pretty high variance.

## update after rebuttal

I remain unenthusiastic about this paper, but I don't strongly object to it.  The main contribution is the proposed parameterization, in terms of the largest coordinate in the spike; given this formulation, getting an algorithm is fairly straightforward.  I do agree that identifying that this is a fairly simple formulation with a potential for significantly improved sample complexity is a real contribution.  But I'm not excited by *further* restrictions on the spiked covariance model, which is already toy.

**Claims And Evidence:**

The paper shows how to achieve O(k log p) sample complexity, which is in general optimal, under an assumption that the largest single coordinate has fairly large variance.

**Essential References Not Discussed:**

Not that I know of.

**Experimental Designs Or Analyses:**

The experiments are just synthetic, presumably because the spiked covariance model is a bit of a toy.

**Methods And Evaluation Criteria:**

Under the assumption given, the result seems like it should be pretty
straightforward to achieve.  Why not just take (Sigma^ e_{j0})_{S^} as
your estimate v^0?

For any j != j0, (Sigma^ e_{j0})_{j} is going to be distributed as

  (1/m) sum_i lambda v_j0 v_j g_i^2 + (N(0, 1) * N(0,1))

The noise term is independent and constant variance, so it sums to
basically N(0, O(m)), and will be O(sqrt(m log p)) for all
coordinates; the leading term has (sum g_i^2) being (m +/- sqrt(m log
p)) for all coordinates; and by assumption, lambda v_j0 > 1.  So we
get the value

  v^0_j = (lambda v_j) v_j +/- v_j sqrt(log p / m) +/- sqrt(log p / m)

The scaling (lambda v_j) > 1 only helps; ignoring it, we have v_j < 1 so the error is

 |v^0_j - v_j| <~ sqrt(log p / m)

For the given m = (1/gamma^2) k log p, this is coordinatewise error at
most gamma/sqrt(k).  That Linf error of a sparse vector, after
restricting to the largest k entries, gives gamma L2 error.

Am I missing something?

**Other Comments Or Suggestions:**

see above

**Other Strengths And Weaknesses:**

The single-spiked covariance model is already a toy model, so adding another assumption beyond the basics isn't that exciting.

**Questions For Authors:**

Please address my question about a simpler algorithm above.

**Relation To Broader Scientific Literature:**

There's an annoying gap between what we can achieve computationally and information theoretically for sparse PCA; this paper looks at how to work around that gap by parameterizing in terms of a different value.

**Theoretical Claims:**

I didn't check the proofs carefully, but the result seems pretty straightforward so I believe them.

---

> ### Author Rebuttal · Authors · 2025-04-01
>
> > 1. Under the assumption given, the result seems like it should be pretty straightforward to achieve. Why not just take $(\hat{\Sigma} e _{j _0}) _{\hat{S}}$ as your estimate $v^0$? The scaling $(\lambda v_j) > 1$ only helps; ignoring it, we have $v_j < 1$ so the error is $|v^0 _j - v_j| \leq c \sqrt{\log p / m}$.
>
> **Reply:** We thank the reviewer for the interesting suggestion. Our response is organized in three parts.
>
> **1. Slight Modification Required for the Reviewer’s Estimator**
>
> The reviewer suggested using $ (\hat{\Sigma} e _{j _0}) _{\hat{S}} $ as an estimator for $v$, where $e _{j _0}$ is the canonical basis vector. However, under the spiked covariance model, we have $E[\hat{\Sigma}] = \lambda v v^T + I$, so that $E[\hat{\Sigma}] e _{j _0} = \lambda  v _{j _0} v + e _{j _0}$. Thus, while for $j \neq j_0$ the entry is $\lambda v _{j_0} v_j$, the $j_0$-th entry becomes $\lambda v _{j_0}^2 + 1$, introducing a bias of $1$. To remove this bias, we subtract $e _{j_0}$ and define the modified estimator
> $$
> \hat{v} _{\text{new}} = \frac{(\hat{\Sigma} e _{j _0} - e _{j _0}) _{\hat{S}}}{ ||(\hat{\Sigma} e _{j _0} - e _{j _0}) _{\hat{S}} ||_2},
> $$
> thereby properly centering the estimator.
>
> **2. Technical Challenges**
>
> Both the estimator $\hat{v} _{\text{new}}$ and our proposed estimator rely on the random index $j_0$ and the estimated support $\hat{S}$. Because of this, establishing the sample complexity of $\hat{v} _{\text{new}}$ is nontrivial and requires extra technical development.
>
> First, since $\hat{v} _{\text{new}}$ involves $j_0$ and $\hat{S}$, its analysis requires key results (e.g. Lemmas A.6 and A.8) to control their randomness. In particular, Lemma A.8 (see Lines 880--888) directly leads to the assumption $\lambda = \Omega ( ||v|| _{\infty}^{-1} )$, which ensures the necessary probability bounds in concentration inequalities (as in Eq. (23)).
>
> Second, since $j_0$ is random, one cannot treat $\hat{\Sigma} _{j,j_0}$ as if $j_0$ were fixed; similar techniques as in Lemma A.8 are needed to handle its variability.
>
> Third, since the true vector $v$ is unknown, $\lambda v _{j _0}$ cannot directly serve as the scaling between $(\hat{\Sigma} e _{j _0} - e _{j _0}) _{\hat{S}}$ and $v$. Instead, one must first accurately estimate the $L_2$ norm $|| (\hat{\Sigma} e _{j _0} - e _{j _0}) _{\hat{S}} ||_2$
> to properly normalize the estimator.
>
> **3. Comparison: Computational Load and Empirical Performance**
>
> The two estimators differ mainly in their final computational steps. Our proposed estimator (via Algorithm 1) computes the leading eigenvector of the $k \times k$ submatrix $\hat{\Sigma} _{\hat{S}}$, which costs $O(nk^2)$, while the modified estimator $\hat{v} _{\text{new}}$ only normalizes $(\hat{\Sigma} e _{j_0} - e _{j_0}) _{\hat{S}}$, costing $O(p)$. Although $\hat{v} _{\text{new}}$ is more efficient, our experiments show that the estimator via Algorithm 1 achieves lower estimation error. For example, Table 1 displays estimation error versus sample size (with dimension $p = 4000$, sparsity $k = 20$ and signal strength $\lambda = 10$), and Table 2 reports the corresponding computational times.
>
> **Table 1. Estimation error versus sample size for two estimators, with $p = 4000$, $k = 100$, $\lambda = 10$.**
>
> | Sample size             | 500   | 1000    | 1500    | 2000    | 2500    | 3000    | 3500    | 4000    |
> |-------------------------------|--------|--------|--------|--------|--------|--------|--------|--------|
> | Estimator via Algorithm 1     | 0.9593 | 0.6111 | 0.4309 | 0.3104 | 0.2274 | 0.1652 | 0.1272 | 0.0895 |
> | Estimator $\hat{v}_{\text{new}}$     | 1.0920 | 0.7596 | 0.5534 | 0.4136 | 0.3215 | 0.2578 | 0.2215 | 0.1867 |
>
> **Table 2. Computational time(s) versus sample size for two estimators, with $p = 4000$, $k = 100$, $\lambda = 10$.**
>
> | Sample size             | 500   | 1000    | 1500    | 2000   | 2500    | 3000    | 3500    | 4000    |
> |-------------------------------|--------|--------|--------|--------|--------|--------|--------|--------|
> | Estimator via Algorithm 1     | 0.0887 | 0.1192 | 0.1479 | 0.1737 | 0.1907 | 0.2145 | 0.2416 | 0.2648 |
> | Estimator $\hat{v}_{\text{new}}$     | 0.0854 | 0.1168 | 0.1454 | 0.1707 | 0.1875 | 0.2113 | 0.2383 | 0.2622 |
>
> We will add further discussion and experimental evaluations of this estimate in the revised manuscript.
>
> > 2. The single-spiked covariance model is already a toy model, so adding another assumption beyond the basics isn't that exciting.
>
> **Reply:** Although the single-spiked covariance model is a toy model, a significant gap still exists between the information-theoretic sample complexity and what polynomial-time algorithms achieve. Under the planted clique conjecture, it is known that no polynomial-time algorithm can recover the spike at the optimal sample complexity without additional assumptions. **Please refer to the response to the first question for Reviewer gGz9 for a detailed discussion on the necessity of this assumption.**

---

> > ### Comment · Reviewer_LyQJ · 2025-04-05
> >
> > Right, you do need to subtract e_{j0}, thanks for the reminder.  But other than that I don't think these challenges are significant:
> >
> > - j0 being not fixed doesn't really matter -- the concentration is so good, you could just union bound over possible j0.  (So union bound over p^2 not p things -- nbd.)
> >
> > - you don't need to accurately estimate the L2 norm, since you're estimating a unit vector; you'll normalize it anyway.
> >
> > I'll raise my score a bit, but I'm not convinced -- and sure the empirical performance is slightly better, but probably a more thorough local search would further improve it.  Empirical performance isn't really the main selling point here.

---

> > > ### Author Response · Authors · 2025-04-06
> > >
> > > Thank you for your insightful feedback. Below, we try our best to address your concerns and strengthen our contribution by providing additional clarification of our method.
> > >
> > > Our proposed thresholding algorithm comprises three steps:
> > >
> > > 1. Estimate the index corresponding to the largest absolute entry of the true spike by $j_0 = \arg \max_j \hat{\Sigma}_{j,j}$.
> > >
> > > 2. Recover the full support of the true spike by selecting the indices of the top $k$ absolute entries from the $j_0$-th column of $\hat{\Sigma}$.
> > >
> > > 3. Estimate the spike's values within the estimated support via eigendecomposition.
> > >
> > > **The first two steps are critical for achieving the sample complexity of $\Omega(k\log p)$**, as demonstrated in Lemmas~A.6 and A.8. In particular, by amplifying the separation between the in-support and out-of-support entries compared with typical diagonal thresholding (see Equations (5) and (7)), these steps make it easier to distinguish the two sets and require fewer samples to recover the support. This enhanced separation is the innovation of our method. Notably, **alternative procedures for the third step—such as the variant you suggested—can be employed without affecting the overall sample complexity**.
> > >
> > >
> > > By contrast, diagonal thresholding directly selects the top $k$ diagonal entries of $\hat{\Sigma}$ for support recovery. Compared with our approach, this method produces a less pronounced separation between the in-support and out-of-support entries, which in turn results in a higher sample complexity of $\Omega(k^2\log p)$, even though the final step for estimating the spike's values within the support is identical.
> > >
> > > Finally, the necessity of introducing an additional assumption is underscored by reductions from the planted clique conjecture, which strongly suggest that **without extra condition, no polynomial-time algorithm can achieve $\Omega(k\log p)$ sample complexity**.
> > >
> > > We also thank the reviewer for suggesting a variant estimation procedure that offers improved computational efficiency at the expense of some estimation performance. A discussion of this alternative will be included in the final version of the paper.

---

### Decision · Program_Chairs · 2025-05-01

**Decision:**

Accept (poster)

**Comment:**

This paper proposes a two-stage algorithm for sparse PCA under the single-spiked covariance model, achieving near-optimal
$O(k \log p)$ sample complexity by assuming a large leading coordinate in the spike. The method combines a thresholding-based support recovery step with truncated power iteration for refinement.

The contribution is technically solid and closes a known computational-statistical gap under a plausible structural assumption. While the estimator is conceptually simple, the authors justify its analysis and show improved empirical accuracy compared to baseline variants. The rebuttal clarified several points, including robustness to the assumption, diagonal thresholding comparisons, and the role of the refinement step. The main limitation is that the setting is still a simplified model and results are only shown on synthetic data. The extra assumption helps the theory but may not always hold in real problems. However, given the strong theoretical contribution, I recommend an accept.